# Highly biased agonism for GPCR ligands via nanobody tethering

Shivani Sachdev[1], Brendan A. Creemer[1], Thomas J. Gardella[2] & Ross W. Cheloha [1] ✉

Ligand-induced activation of G protein-coupled receptors (GPCRs) can initiate signaling through multiple distinct pathways with differing biological and physiological outcomes. There is intense interest in understanding how variation in GPCR ligand structure can be used to promote pathway selective signaling ("biased agonism") with the goal of promoting desirable responses and avoiding deleterious side effects. Here we present an approach in which a conventional peptide ligand for the type 1 parathyroid hormone receptor (PTHR1) is converted from an agonist which induces signaling through all relevant pathways to a compound that is highly selective for a single pathway. This is achieved not through variation in the core structure of the agonist, but rather by linking it to a nanobody tethering agent that binds with high affinity to a separate site on the receptor not involved in signal transduction. The resulting conjugate represents the most biased agonist of PTHR1 reported to date. This approach holds promise for facile generation of pathway selective ligands for other GPCRs.

G protein-coupled receptors (GPCRs) are the largest family of cell surface proteins and the most common target of approved therapeutics. These receptors are activated by diverse molecules and stimuli ranging from proteins and peptides to small molecules and protons[1]. Recent structural studies using X-ray crystallography and cryo-electron microscopy have revealed mechanisms of receptor activation[2,3]. A handful of common signatures have been observed in structures of activated receptors in complex with G proteins or ligands relative to unactivated counterparts[4]. Even among the receptor structures classified as "active state", subtle structural differences are observed among receptors bound to different ligands, G-proteins, and accessory proteins[5].

A wide body of experimental findings has demonstrated that different compounds that bind to and activate the same receptor induce divergent biological and physiological responses in a paradigm known as ligand bias or pathway selective signaling[6,7]. Biased ligands induce preferential signaling through one receptor transducer (such as Gαs-mediated adenylate cyclase activation) over another (such as β-arrestin) relative to an index comparator ligand[8]. Extensive effort has

been invested to link the subtle differences in receptor structure induced by the binding of different ligands to biased signaling outputs[9]. In some examples, there is a clear connection between variation in the receptor conformation induced by the binding of a biased ligand and the type of signaling bias observed, though this is not true for all examples[10,11]. As such, prospective efforts for the rational design of biased agonists with desired signaling bias profiles are challenging.

The GPCR superfamily is divided into separate classes based on sequence conservation and the presence of class-specific structural features[12]. Class B1 GPCRs are characterized by a large extracellular domain (ECD), which facilitates the binding of medium-sized polypeptide ligands[2]. The interaction between class B1 GPCRs and their ligands has traditionally been described as a two-site mode of binding[13]. In this two-site mechanism, the interaction between the C-terminal portion of the peptide ligand and the ECD of the receptor (site 1 interaction) provides most of the binding affinity and specificity for the bimolecular interaction. Site 1 also serves to anchor the N-terminal portion of the peptide in a position where it can contact the transmembrane portion of the receptor (site 2 interaction, orthosteric

[1]Laboratory of Bioorganic Chemistry, National Institutes of Diabetes, Digestive and Kidney Diseases, National Institutes of Health, Bathesda, MD, USA.
[2]Endocrine Unit, Massachusetts General Hospital and Harvard Medical School, Boston, MA, USA. ✉e-mail: ross.cheloha@nih.gov

site), induce a conformational change, and initiate a signaling response. Although the site 2 interaction is low affinity, it is hypothesized to be singularly responsible for ligand agonist behavior, including the induction of pathway selective signaling[14–18]. In the two-site model, the ECD-peptide C-terminus interaction serves as a tethering point to improve ligand affinity and potency but plays no direct role in inducing a conformational change. By extension, structural modifications in the ligand C-terminus are predicted to substantially impact ligand binding but not pathway-selective signaling, although recent findings suggest this is an oversimplification[19,20].

The interaction of the type-1 parathyroid hormone receptor (PTHR1), a class B1 GPCR, with its ligands has served as an exemplar of this two-site mechanism of ligand binding. PTHR1 is bound by two naturally occurring peptide hormones, parathyroid hormone (PTH) and PTH-related protein (PTHrP), with full biological activity residing in the N-terminal 34 or 36 residues, respectively[21]. Activation of PTHR1 induces signaling through $G\alpha$ proteins (primarily $G\alpha s$ and secondarily $G\alpha q$) and $\beta$-arrestin. Modifications in the C-terminal portion of $PTH_{1-34}$ have strong effects on receptor binding and signaling potency (site 1), whereas N-terminal modifications mostly impact receptor activation efficacy and pathway selectivity (site 2). For example, modifications at positions 23, 24, and 28 in $PTH_{1-34}$ diminish binding to the receptor ECD and signaling potency without a substantial impact on ligand efficacy or pathway selectivity[16,22,23]. In contrast, modifications within the first 11 residues of PTH have been identified that affect ligand agonist efficacy, pathway-signaling selectivity, and signaling localization[16–18,24,25].

Some optimized $PTH_{1-11}$ analogs exhibit signaling profiles and pathway selectivity similar to $PTH_{1-34}$, albeit with somewhat reduced potency and affinity[24,26]. Past work showed that such truncated PTH analogs could be linked to single domain antibodies (nanobodies, Nbs) using a combination of enzymatic protein labeling, solid phase peptide synthesis, and chemoselective conjugation chemistry[27]. This approach, termed "CLAMP", uses Nbs, which are the smallest antibody fragments that maintain the desirable characteristics of conventional antibodies, such as high target affinity and specificity[28]. In contrast to conventional antibodies, Nbs can be produced in high yield from bacteria, exhibit high stability even in the absence of glycosylation and disulfide bond formation, and are comprised of a single polypeptide chain, which facilitates straightforward construction of multi-specific conjugates. Past work has shown that the linkage of a $PTH_{1-11}$ analog with an Nb that bound to the PTHR1 extracellular domain augmented signaling through the $G\alpha s$ pathway[27]. The binding of Nb to ECD can be envisioned as a surrogate for the site 1 interaction engaged by the $PTH_{1-34}$ C-terminal region. The conventional two-site model would suggest that $PTH_{1-11}$-Nb conjugates would exhibit agonist properties (signaling efficacy, pathway selectivity, and signaling localization) similar to $PTH_{1-11}$ alone.

Here we test this hypothesis by synthesizing a variety of Nb-$PTH_{1-11}$ conjugates through chemical, enzymatic, and recombinant protein expression methods. Using a panel of Nbs, including a PTHR1-binding Nb ($Nb_{PTHR1-X2}$) characterized herein, we show that $PTH_{1-11}$-Nb conjugates are unexpectedly highly selective for $G\alpha s$/cAMP pathway activation. This finding is in stark contrast to conventional PTHR1 ligands, such as $PTH_{1-11}$ and $PTH_{1-34}$, which signal through all PTHR1-engaged pathways. Mechanistic studies revealed that the $PTH_{1-11}$-Nb conjugates can activate signaling through a mode that involves two receptor protomers (termed "activation in trans"), which is qualitatively distinct from that of conventional ligands of PTHR1. These findings show that agonist properties, such as pathway selectivity for class B1 GPCRs, are dependent not only on the structure of the ligand that engages the orthosteric site (site 2) but also on the positioning of the high affinity tethering interaction (site 1). We further extend this work by showing that an analogous approach can be used to generate ligand-Nb conjugates that target glucagon-like peptide receptor-1

(GLP1R) and exhibit selective signaling through the $G\alpha s$/cAMP pathway. This finding holds important implications for efforts to develop potent-biased agonists.

## Results

We sought to probe the consequences of outsourcing the receptor-binding function of PTHR1 ligands to artificial building blocks and non-natural binding sites. Towards this end, we synthesized a set of ligands designed to target either WT PTHR1 or engineered receptors (Fig. 1). Nb–ligand conjugates were prepared using recombinant Nb expression, site-specific labeling, peptide synthesis, and click chemistry (Fig. 1B). All peptides, including $PTH_{1-11}$-6E, which contains a six-carbon linker (6-aminohexanoic acid, Ahx) between components, were synthesized using standard Fmoc-based solid phase peptide synthesis. Peptide and conjugate identity were confirmed by mass spectrometry (Supplementary Tables 1 and 2). Previously described variants of PTHR1 (Fig. 1C) engineered to contain a high-affinity binding site for an epitope tag-binding Nb ($Nb_{6E}$ binds PTHR1-6E) or an epitope peptide (6E peptide binds PTHR1-$Nb_{6E}$) were also deployed[27,29]. Either epitope tag (6E) or Nb ($Nb_{6E}$) were engrafted into a flexible and unstructured region of the receptor encoded by exon 2, which is known to be dispensable for high-affinity ligand binding[30]. Through this design, the signaling properties of $PTH_{1-34}$ could be compared to engineered ligands. For example, both $PTH_{1-34}$ and $PTH_{1-11}$-6E are predicted to engage in a high-affinity interaction with PTHR1-$Nb_{6E}$ but with different mechanisms (Fig. 1C). $PTH_{1-34}$ affinity comes mostly through engagement of a cleft in the receptor extracellular domain by the $PTH_{12-34}$ fragment, whereas $PTH_{1-11}$-6E instead binds to the engrafted $Nb_{6E}$. Analogously, both $PTH_{1-11}$- $Nb_{6E}$ and $PTH_{1-34}$ were predicted to exhibit high-affinity binding at PTHR1-6E.

Experiments were performed to assess how signaling characteristics varied in response to alterations in the mode of interaction between the receptor and ligand conjugates. PTHR1 signals primarily through the $G\alpha s$ pathway, which induces intracellular cyclic adenosine monophosphate (cAMP) production. Previously developed cell lines, derived from HEK293, which stably express individual PTHR1 variants of interest, and a luciferase-based biosensor used to monitor cyclic adenosine monophosphate (cAMP) production were applied[27,31]. We compared the activities of $PTH_{1-34}$, $PTH_{1-11}$, and $PTH_{1-11}$-conjugates in cell lines expressing either PTHR1-6E or PTHR1-$Nb_{6E}$. For both cell lines, $PTH_{1-34}$ exhibited a typical PTHR1 cAMP dose–response pattern (Fig. 2, Supplementary Fig. 1), with $EC_{50}$ values in the low nM range (Table 1). As expected, the $PTH_{1-11}$ peptide fragment, which lacks residues important for ECD binding, showed a reduced potency compared to full-length $PTH_{1-34}$. Conjugation of $PTH_{1-11}$ with $Nb_{6E}$, which binds to the 6E epitope in the ECD of PTHR1-6E, caused a substantial enhancement in potency in cells expressing this receptor (Fig. 2, Table 1), in agreement with past findings[27]. $PTH_{1-11}$-$Nb_{6E}$ conjugates were inactive on cells expressing PTHR1 lacking this epitope tag (Supplementary Fig. 2), in agreement with past results. The conjugation of $PTH_{1-11}$ to a negative control nanobody ($PTH_{1-11}$-$Nb_{Neg}$), which recognizes an epitope not present (BC2)[32], provided conjugates with no activity at PTHR1-6E (Fig. 2). We also evaluated the persistence of cAMP generation for conjugates following removal of free ligand ("washout"). Past work has shown that ligands that exhibit prolonged PTHR1 washout responses also induce prolonged physiological activity in vivo[20]. Our data reveal that $PTH_{1-11}$-$Nb_{6E}$-induced cAMP responses were similar in magnitude and duration to $PTH_{1-34}$, remaining stable throughout the course of 30 min (Fig. 2). Signaling duration was also evaluated through measurement of the area under the curve (AUC) after washout, also in agreement with past findings[27]. By this measure, $PTH_{1-11}$-$Nb_{6E}$ induced more enduring washout responses than $PTH_{1-34}$, whereas responses from $PTH_{1-11}$ were weaker (Fig. 2, Table 1).

Similar experiments were performed on the cell line expressing PTHR1-$Nb_{6E}$ (Fig. 2, Supplementary Fig. 3). In this context, $PTH_{1-11}$-6E

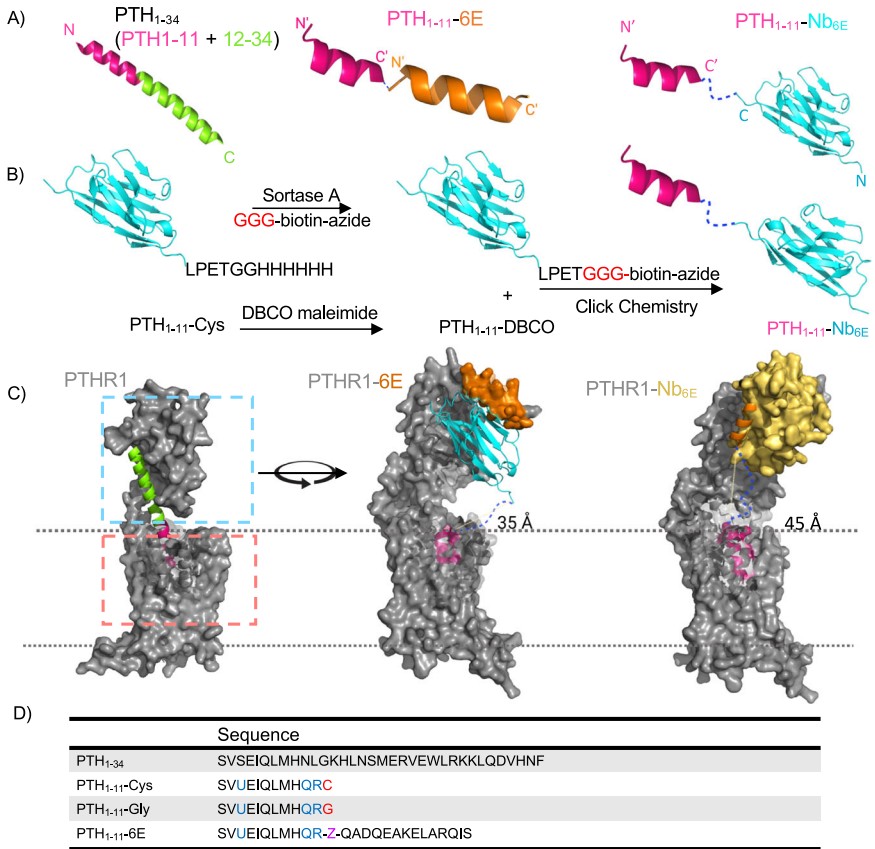

**Fig. 1 | Ligand–nanobody conjugate synthesis and receptor designs.**
**A** Schematic representation of the ligands used in this study. **B** Synthetic scheme for the preparation of Nb–ligand conjugates. **C** Variation on the canonical two-site binding model of Class B1 GPCR activation. For PTHR1, the model is based on the published crystal structure of $PTH_{1-34}$ bound to PTHR1 (PDB: 6FJ3). The peptide agonist–ECD interaction is highlighted in the blue dashed box (site 1), and the agonist–transmembrane domain interaction is shown in the salmon dashed box (site 2). Models of PTHR1-6E and $PTHR1-Nb_{6E}$ interacting with binding partners were generated using Alphafold2 (see the "Methods" section) and are shown rotated relative to wild-type PTHR1. PTHR1-6E is shown bound to $PTH_{1-11}-Nb_{6E}$ conjugate (pink and cyan). $PTHR1-Nb_{6E}$ is shown bound $PTH_{1-11}-6E$ (pink and orange). The dark blue dashed lines depict a hypothetical linker between the C-terminus of $PTH_{1-11}$ and $Nb_{6E}$ or 6E in these models. Distances listed were measured in Alphafold generated models using Pymol. **D** Sequences of ligands used in this study. Sites with mutations relative to natural $PTH_{1-34}$ are highlighted with colored text. The residue abbreviated with "U" corresponds aminoisobutyric acid; "Z" corresponds to 6-aminohexanoic acid. Mass spectrometry characterization of compounds is shown in Supplementary Tables 1 and 2. Unless stated otherwise, reference to "$PTH_{1-11}$" in the remainder of the text refers to $PTH_{1-11}$-Gly, whereas Nb-$PTH_{1-11}$ conjugates are constructed using $PTH_{1-11}$-Cys.

more effectively induced cAMP production than $PTH_{1-11}$, with a 400-fold difference in potency (Table 1). In contrast, $PTH_{1-11}$-6E was no more active than $PTH_{1-11}$ in cells expressing PTHR1 without $Nb_{6E}$ engrafted (Supplementary Fig. 2). In washout assays, $PTH_{1-11}$-6E induced more enduring cAMP responses relative to the other peptides tested. To assess whether the prolonged signaling from $PTH_{1-11}$-6E corresponded with continuous engagement of receptor or whether two-site binding permitted repeated association/dissociation cycles at different receptor sites, we evaluated the impact of antagonists added at the beginning of the washout phase (Supplementary Fig. 4). $PTH_{1-11}$-6E washout responses were highly sensitive to inhibitors. The addition of synthetic 6E peptides causes inhibition through competition with $PTH_{1-11}$-6E for $Nb_{6E}$ binding, whereas SW106 interferes with the binding of $PTH_{1-11}$[33]. The marked effects of both types of inhibitors on $PTH_{1-11}$-6E signaling suggest that this ligand does not continuously engage both of its binding sites throughout washout. The synthetic 6E peptide competitor is cell impermeable; its effective antagonism in this context suggests much of the signaling of $PTH_{1-11}$-6E originates from the cell surface. Similarly high levels of inhibitor sensitivity were previously observed for analogs of $PTH_{1-34}$ that specifically signal from the cell surface[18,25].

To support these findings, we conducted a distinct assay, based on the use of a bioluminescence resonance energy transfer (BRET)

biosensor[34], to measure ligand-induced Gαs dissociation from the plasma membrane. This experimental format does not rely on the signal amplification inherent in the cAMP-based assay described above. In this assay, $PTH_{1-34}$ exhibits a potency similar to that previously reported in experiments using wild-type PTHR1[34]. $PTH_{1-11}-Nb_{6E}$ or $PTH_{1-11}$-6E exhibited potency and efficacy comparable to $PTH_{1-34}$ on PTHR1-6E or $PTHR1-Nb_{6E}$, respectively (Fig. 3A), in line with their behavior in the cAMP assay. The observation of higher potency for PTHR1 ligands in the Gαs BRET assay in comparison to the cAMP reporter assay conforms with previous reports[34,35]

We also measured ligand-induced recruitment of β-arrestin 2 to the plasma membrane and to early endosomes using a bystander BRET assay[34]. In each assay, $PTH_{1-34}$ showed the expected dose–response for induction of β-arrestin 2 recruitment (Fig. 3B, C, Table 1), with somewhat diminished potency compared to cAMP assays. $PTH_{1-11}$ was less potent relative to $PTH_{1-34}$. Strikingly, $PTH_{1-11}-Nb_{6E}$ displayed a drastically impaired capacity to induce recruitment of β-arrestin 2 to the plasma membrane or to early endosomes in cells stably expressing PTHR1-6E (Fig. 3, Supplementary Fig. 5). Assays in cells expressing $PTHR1-Nb_{6E}$ showed that $PTH_{1-11}$-6E was nearly inactive for inducing β-arrestin 2 recruitment, even at a concentration as high as 30 μM (Fig. 3B, C). A comparable lack of activity was recorded for $PTH_{1-11}-Nb_{6E}$ and $PTH_{1-11}$-6E for inducing Gαq-mediated calcium mobilization in

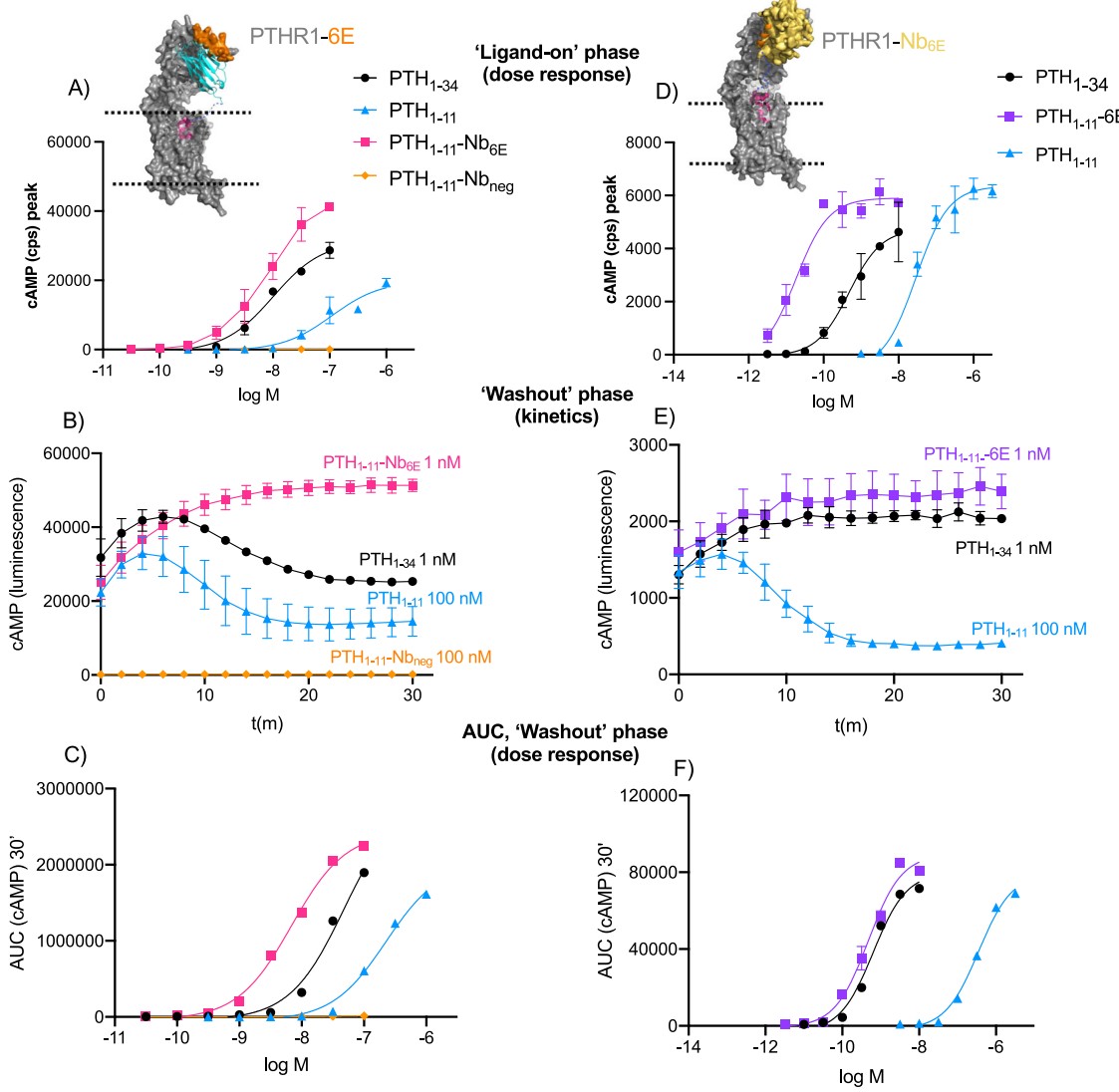

**Fig. 2 | Ligand-induced cAMP responses of engineered receptors.** Data correspond to experiments run using cells expressing PTHR1-6E (panels **A**–**C**) or PTHR1-Nb_{6E} (panels **D**–**F**). **A** Concentration–response curve for cAMP production in cells expressing PTHR1-6E. Consistent color coding for ligands is used throughout the figure. *Y*-axes refer to maximal cAMP responses observed in a plate reader, quantified by counts per second "cps". **B** Representative kinetic plots for ligand-induced signaling following the removal of the unbound ligand in a washout assay. **C** Quantitation of the duration of washout responses summarized as AUC. Panels **D**–**F** show analogous data for cells expressing PTHR1-Nb_{6E}. Data points correspond to mean and associated SD from technical duplicates in a representative experiment. Curves were generated using a three-parameter logistic sigmoidal model. Tabulation of agonist potency parameters is shown in Table 1, which are derived from 3 to 5 independent experiments with the precise number of biological replicates shown in Supplementary Table 3. Independent replicates are shown in Supplementary Figs. 1 and 3. Source Data used to generate graphs are provided in Source Data Files.

relevant cell lines (Fig. 3C). The capacity of PTH_{1-11}-Nb_{6E} and PTH_{1-11}-6E to robustly induce cAMP responses in relevant cell lines, without concomitant induction of β-arrestin 2 recruitment or Gαq activation, corresponds to a clearcut example of highly biased agonism.

This finding prompted us to investigate whether the biased agonism profile observed with engineered receptors would translate to the native human PTHR1 receptor (hPTHR1). This required the use of Nbs that bind directly to hPTHR1. Sequences for Nbs that recognize PTHR1 (named here Nb_{PTHR1} and Nb_{PTHR1-X2}) have been reported[36], albeit with sparse characterization. Nb_{PTHR1} was previously used (with the name VHH_{PTHR}) for ligand tethering studies[27], whereas Nb_{PTHR1-X2} is uncharacterized in this context. Unlike the experiments above, the binding sites of these Nbs are not defined by the location of engineered tags, and they have not been determined experimentally. To elucidate the topology of conjugate–receptor interactions, we sought to characterize the epitopes of Nb_{PTHR1} and Nb_{PTHR1-X2} on the receptor.

We labeled Nb_{PTHR1} and Nb_{PTHR1-X2} with detection tags via sortagging and analyzed binding to hPTHR1 stably expressed on HEK293 cells with flow cytometry. Both Nbs bound to hPTHR1 but only Nb_{PTHR1} bound to PTHR1-6E, suggesting the binding site of Nb_{PTHR1-X2} lies within the region of PTHR1 replaced by the 6E epitope, found within the disordered exon 2-encoded region of the receptor (Fig. 4A, Supplementary Fig. 6A). This finding was corroborated with signaling assays in cells expressing PTHR1-6E (Supplementary Fig. 6) and rat PTHR1[37] (Supplementary Fig. 7) on which Nb_{PTHR1-X2} conjugates were inactive. Addition of an exogenous synthetic peptide corresponding to the putative epitope of Nb_{PTHR1-X2} blocked the engagement of receptor by Nb_{PTHR1-X2}, but not Nb_{PTHR1} (Supplementary Fig. 8). The effective blockade of Nb_{PTHR1-X2} binding to the receptor by a short peptide suggests that epitope folding is likely not a major determinant of Nb recognition. This aligns with previous work demonstrating that the portion of PTHR1 encompassing this epitope is dynamic and not easily

**Table 1 | Pharmacological parameters for ligands and conjugates across all the functional assays tested**

| $EC_{50}$ (±SEM) Ligands | cAMP | Washout AUC | Gαs | β-arrestin 2 (plasma membrane) | β-arrestin 2 (endosome) |
|---|---|---|---|---|---|
| **PTHR1-6E** | | | | | |
| $PTH_{1-34}$ | 25 (8) | 36 (8) | 0.3 (0.1) | 189 (48) | 142 (62) |
| $PTH_{1-11}$ | 50 (23) | 125 (57) | 32 (13) | 11300 (3100) | 4800 (900) |
| $PTH_{1-11}$-$Nb_{6E}$ | 22 (7) | 8 (0.7) | 5 (0.7) | - | - |
| $PTH_{1-11}$-$Nb_{neg}$ | - | - | - | - | - |
| **PTHR1-Nb$_{6E}$** | | | | | |
| $PTH_{1-34}$ | 0.4 (0.1) | 1.2 (0.4) | 1.1 (0.8) | 106 (72) | 54 (19) |
| $PTH_{1-11}$ | 28 (11) | 240 (72) | 198 (142) | 27200 (7200) | 23100 (1400) |
| $PTH_{1-11}$-6E | 0.07 | 0.2 (0.08) | 4.4 (3) | - | - |
| **PTHR1** | | | | | |
| $PTH_{1-34}$ | 5 (2) | 5.4 (1.7) | 0.1 (0.06) | 62 (12) | 25 (3) |
| $PTH_{1-11}$ | 22 (5) | 342 (116) | 26 (2.6) | 12300 (1700) | 11900 (4200) |
| $PTH_{1-11}$-$Nb_{PTHR1}$ | 3 (0.4) | 4 (0.6) | 2.3 (0.9) | - | - |
| $PTH_{1-11}$-$Nb_{PTHR1X2}$ | 5.3 (1.5) | 2.3 (1) | ND | - | ND |

Values shown corresponds to mean (SEM) from the number of replicates indicated in Supplementary Table 3. The mean potency values are presented in nM. Dashes ("–") indicate that the $EC_{50}$ value could not be determined due to weak ligand activity in that assay. "ND" indicates that the assay was not conducted. Relevant concentration–response graphs are shown in Figs. 2, 3, and 5 and corresponding Supplementary figures.

characterized in structural studies[2]. A binding assay performed with $Nb_{PTHR1}$ and $Nb_{PTHR1}$-X2 equipped with distinct labels demonstrated double labeling, further indicating that $Nb_{PTHR1}$ and $Nb_{PTHR1}$-X2 bind to separate epitopes (Supplementary Fig. 9).

To further define the binding site of $Nb_{PTHR1}$, we used a BRET-based binding assay[38]. The affinity of tetramethylrhodamine (TMR)-labeled $Nb_{PTHR1}$ for nanoluciferase-PTHR1 fusion (nLuc-PTHR1), in which nLuc was fused to the receptor N-terminus, was measured using BRET. $Nb_{PTHR1}$-TMR exhibited modest affinity but robust labeling (Fig. 4B, C, Supplementary Fig. 10), while $PTH_{1-34}$-TMR showed somewhat stronger binding (Supplementary Fig. 11). We developed a competition binding assay on cells expressing nLuc-PTHR1 with $Nb_{PTHR1}$-TMR and unlabeled competitor ligands (Fig. 4C, D, Supplementary Fig. 12). We found that $PTH_{1-34}$, $PTH_{1-28}$, and $PTHrP_{7-36}$ competed with TMR-$Nb_{PTHR1}$ for binding, whereas more truncated ligands such as $PTH_{1-21}$ did not, suggesting that the binding site of $Nb_{PTHR1}$ overlaps with that of residues 21–28 of $PTH_{1-34}$. $Nb_{PTHR1}$-X2 also failed to compete $Nb_{PTHR1}$-TMR for binding, providing further support for their separate binding sites (Supplementary Figs. 12 and 13). Assessment of $Nb_{PTHR1}$ binding to PTHR1 extracellular domain (ECD) using surface plasmon resonance demonstrated high-affinity binding ($K_d$ ~ 1 nM) with slow dissociation rates (Supplementary Fig. 14 and Supplementary Methods).

Characterization of the binding site of $Nb_{PTHR1}$ using BRET binding assays was corroborated by analysis of the interaction of $Nb_{PTHR1}$ with purified PTHR1 (ECD) using hydroxy radical-based footprinting analysis based on plasma-induced modification of biomolecules (PLIMB)[39]. PTHR1 ECD was expressed, refolded, and purified according to past work (see Supplementary Fig. 15 for the protein sequence)[20]. Incubation of PTHR1 ECD with $Nb_{PTHR1}$ resulted in a modest blockade of modification of ECD at three residues (Fig. 4E, Supplementary Fig. 16), two of which (Y134, Y136) are found near the binding site of residues 20-24 of PTH in the PTH-PTHR1 ECD structure (Fig. 4F)[40]. Although these differences in hydroxy radical labeling did not reach statistical significance ($p$ ~ 0.10), they conform with predictions made from competition binding assays. Together, these data support a model in which $Nb_{PTHR1}$ and $Nb_{PTHR1}$-X2 bind at distinct sites on PTHR1 ECD, with the binding site of $Nb_{PTHR1}$ overlapping in part with the binding site of residues 21-28 of $PTH_{1-34}$.

Conjugates produced from $Nb_{PTHR1}$ and $Nb_{PTHR1}$-X2 with $PTH_{1-11}$ were tested on cells expressing hPTHR1 in the panel of assays described for engineered receptors above (Fig. 5A, Supplementary Fig. 17). In Gαs-cAMP assay, the conjugation of $PTH_{1-11}$ to $Nb_{PTHR1}$ or $Nb_{PTHR1}$-X2 increased its potency by 10- to 100-fold (Fig. 5B). Additionally, both $PTH_{1-11}$-$Nb_{PTHR1}$ and -$Nb_{PTHR1}$-X2 conjugates induced substantially

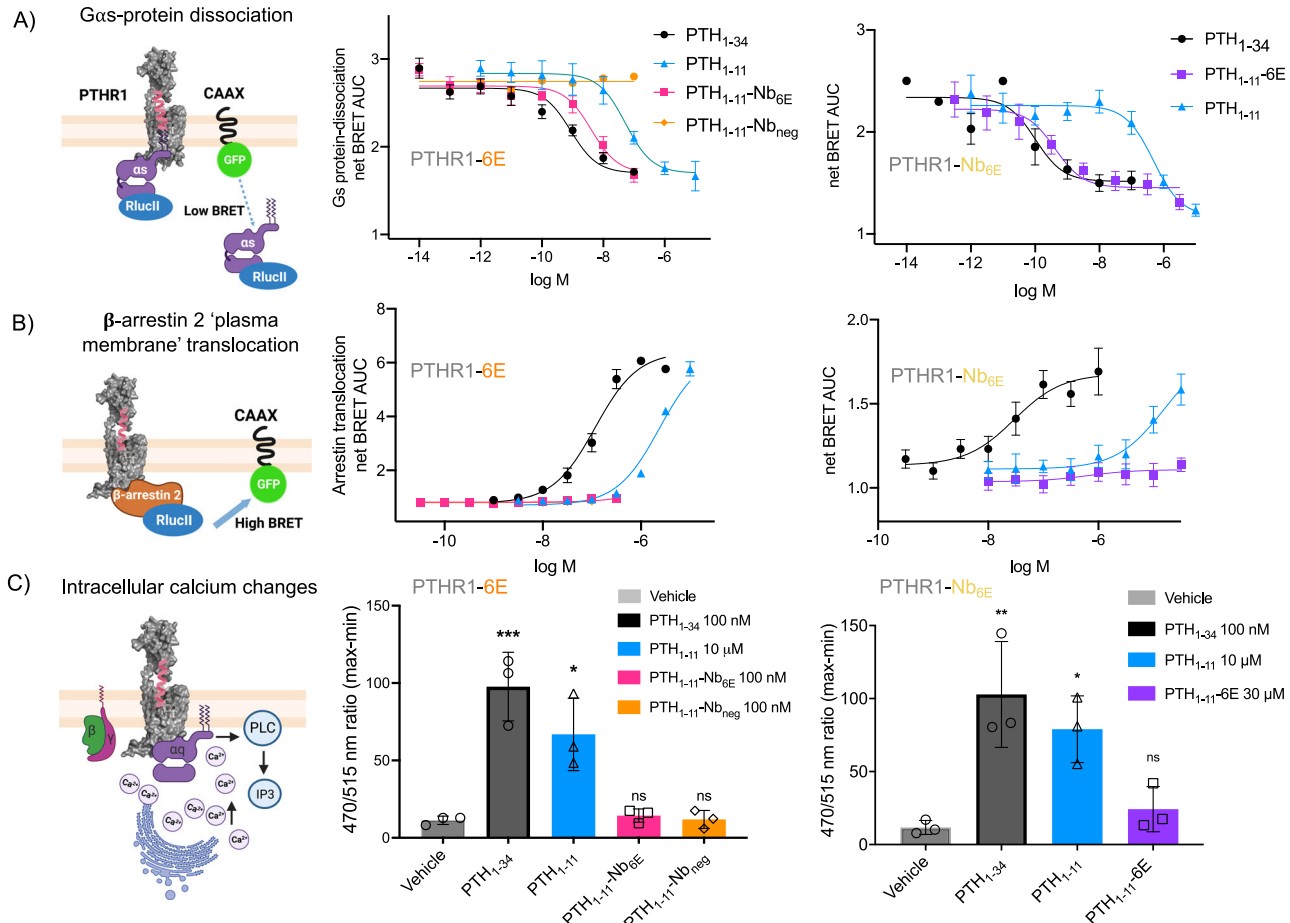

**Fig. 3 | Ligand-induced signaling in engineered receptors through diverse pathways. A** Schematic and data corresponding to a BRET assay used to monitor Gαs-activation. A decrease in the BRET ratio corresponds to ligand-induced dissociation of heterotrimeric Gαs protein complex from the plasma membrane. **B** Schematic and data corresponding to a BRET assay used to monitor β-arrestin 2 recruitment. An increase in the BRET ratio indicates ligand-induced arrestin translocation to the plasma membrane. In panels **A** and **B**, the data correspond to AUC values generated from kinetic measurements. Data points correspond to mean ± SD from technical duplicates in a representative experiment. Characterization of endosomal β-arrestin 2 recruitment is shown in Supplementary Fig. 2. **C** Schematic and data corresponding to the assay used to measure calcium mobilization induced by engagement of the Gαq-PTHR1 pathway. Cells were incubated with the Calbryte 520 AM $Ca^{2+}$ indicator dye and exposed to indicated saturating concentrations of ligands. Data are presented as the relative fluorescence intensity

normalized to the signal background as means ± SEM from 3 to 5 biological replicates, conducted with six technical replicates per experiment. The number of biological replicates for each condition is shown in Supplementary Table 3. Statistical significance was assessed by one-way ANOVA, with Dunnett's post hoc correction (*$p < 0.05$; **$p < 0.01$; ***$p < 0.001$; ****$p < 0.0001$; ns not significant). For PTHR1-6E, Vehicle versus $PTH_{1-34}$ $p = 0.0008$, and Vehicle versus $PTH_{1-11}$ $p = 0.014$. For PTHR1-$Nb_{6E}$, Vehicle versus $PTH_{1-34}$ $p = 0.002$, and Vehicle versus $PTH_{1-11}$ $p = 0.015$. Compiled and quantified data from this figure are shown in Table 1. Independent replicate data are shown in Supplementary Figs. 1 and 3. Source Data used to generate graphs are provided in Source Data Files. This figure was created in part with BioRender.com released under a Creative Commons Attribution-NonCommercial-NoDerivs 4.0 International license: https://creativecommons.org/licenses/by-nc-nd/4.0/deed.en.

longer durations of cAMP production relative to $PTH_{1-11}$ (Fig. 5C, D). Given the rapid decrease of binding as measured by BRET for TMR-$Nb_{PTHR1}$ caused by competitors such as $PTHrP_{7-36}$ (Supplementary Fig. 12), we sought to understand how this observation related to the prolonged cAMP signaling responses induced by conjugates such as $PTH_{1-11}$-$Nb_{PTHR1}$. Building on an assay previously developed to specifically perturb PTHR1 signaling from the cell surface[25], we assessed the impact of adding a peptide antagonist ($PTHrP_{7-36}$) during the washout phase of cAMP assays (Supplementary Fig. 18). We observed that this antagonist had a minimal impact on $PTH_{1-34}$ washout kinetics, in accord with previous findings. In contrast, the addition of $PTHrP_{7-36}$ caused a rapid washout of $PTH_{1-11}$-$Nb_{PTHR1}$ cAMP signaling responses (Supplementary Fig. 18). This observation parallels findings with $PTH_{1-11}$-6E peptides (Supplementary Fig. 4) and suggests that $PTH_{1-11}$-$Nb_{PTHR1}$ conjugates signal mostly from the cell surface and may engage in cycles of binding, dissociation, and reassociation enabled by a multi-site binding mechanism. Another non-exclusive possibility is that

$PTHrP_{7-36}$ acts allosterically to induce $PTH_{1-11}$-$Nb_{PTHR1}$ dissociation, potentially explaining the discrepancy between the high-affinity binding recorded by surface plasmon resonance (Supplementary Fig. 14) and the rapid rate of cAMP signal washout for $PTH_{1-11}$-$Nb_{PTHR1}$ upon addition of $PTHrP_{7-36}$.

Consistent with the ability of $PTH_{1-11}$-Nb conjugates to activate PTHR1 signaling through the Gαs pathway, they promoted Gαs dissociation from the plasma membrane assay with potencies that mirrored those of the cAMP assay (Table 1, Supplementary Fig. 19). Analogously to experiments with engineered receptors, both $PTH_{1-11}$-$Nb_{PTHR1}$ and -$Nb_{PTHR1-X2}$ conjugates displayed negligible recruitment of β-arrestin 2 to the plasma membrane or early endosomes in cells stably expressing hPTHR1 (Fig. 5D, Supplementary Fig. 19). Due to solubility constraints, we were unable to assess concentrations of $PTH_{1-11}$-Nb conjugates above 300 nM, so we cannot exclude the possibility that higher concentrations would induce β-arrestin recruitment. However, we note that $PTH_{1-11}$-$Nb_{PTHR1}$ induced maximal cAMP responses at

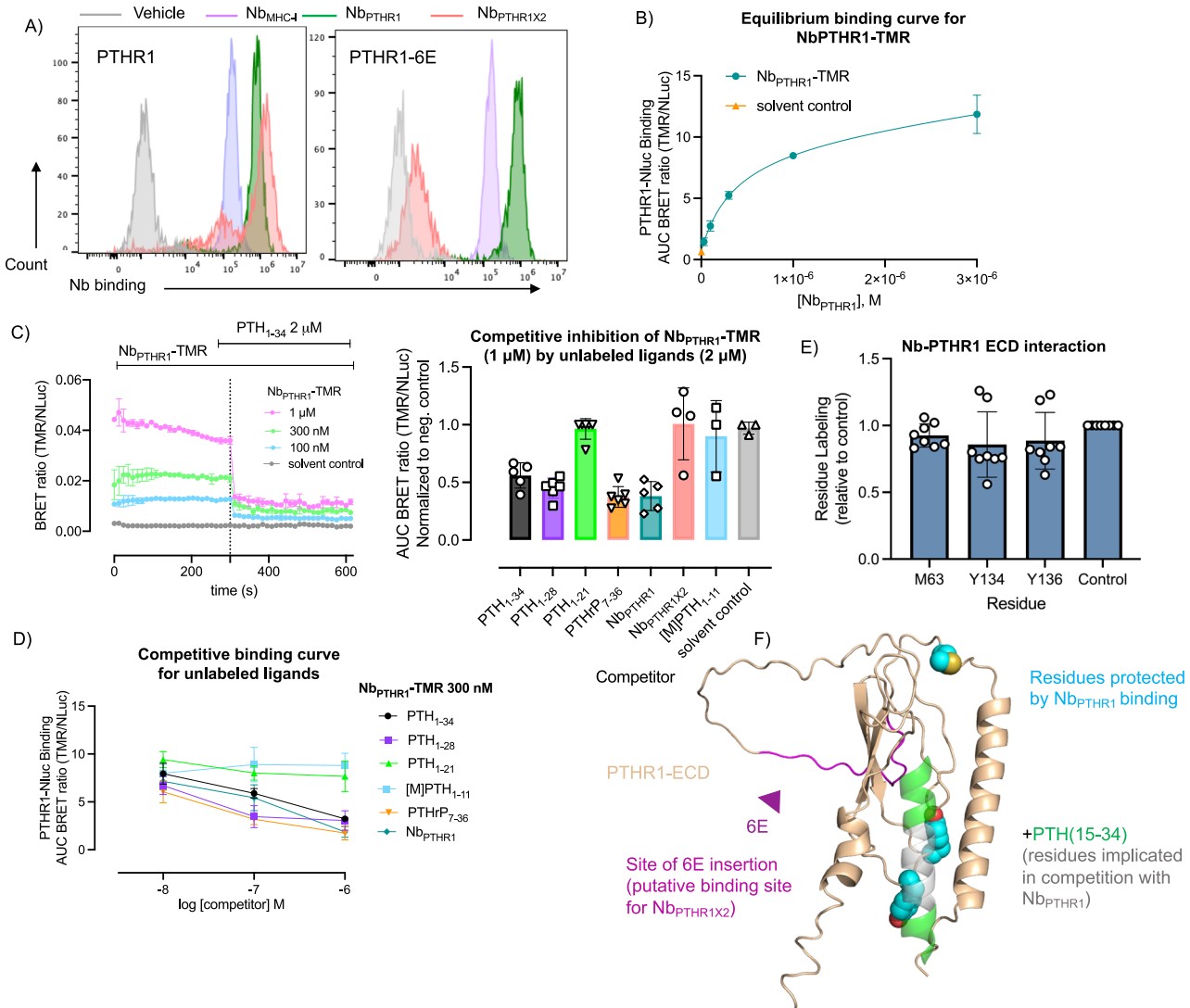

**Fig. 4 | Evaluation of Nb_PTHR1 and Nb_PTHR1X2 binding to WT-PTHR1.**
**A** Representative histograms for flow cytometry analysis of Nb_PTHR1 and Nb_PTHR1-X2 binding to PTHR1 and engineered receptors. Nbs (500 nM) labeled with biotin were incubated with cells expressing PTHR1 or PTHR1-6E, followed by washing, detection with streptavidin-APC, and assessment of cellular fluorescence. Nb_MHC-I (binds MHC-I) is included for comparison to the staining intensity observed with a highly expressed cell surface protein. Analogous data for PTHR1-Nb_6E staining can be found in Supplementary Fig. 6. **B** Representative concentration–response curve for BRET measurements of Nb_PTHR1-TMR binding to cells expressing nLuc-PTHR1. Data points correspond to mean ± SD of AUC measurements, fit to a three-parameter logistic sigmoidal model. Measurements were performed in three independent experiments (Supplementary Fig. 10). **C** (Left) Representative BRET measurements of Nb binding observed upon application of varying concentrations of Nb_PTHR1-TMR followed by the addition of unlabeled competitor peptide (PTH_1-34, 2 μM). (Right) The bar graph shows summarized data for the inhibition of Nb_PTHR1-TMR (1 μM) binding by unlabeled ligands (2 μM). Data correspond to mean ± SEM from independent biological replicates, with each dot corresponding to an independent biological replicate. Quantitation is performed as described in the "Methods"

section. The number of independent biological replicates is listed in parentheses: PTH_1-34(5); PTH_1-28(6); PTH_1-21(4); PTHrP_7-36(6); Nb_PTHR1(5); Nb_PTHR1X2 (4); [M] PTH_1-11(3); solvent control (3). **D** Concentration–response competition binding assays were performed using varying concentrations of unlabeled ligand added simultaneously with Nb_PTHR1-TMR (300 nM). Data points correspond to mean ± SEM from three independent experiments. **E** Analysis of Nb_PTHR1-PTHR1 ECD interactions using hydroxy radical-based footprinting analysis (see the "Methods" section). Only the residues that show a trend towards protection from radical labeling upon the addition of Nb_PTHR1 are shown here, with full data shown in Supplementary Fig. 16. Data corresponds to mean values and associated standard deviation from eight independent replicates. Labeling is normalized to a control performed in the presence of a non-binding Nb. **F** Alphafold2 model of PTHR1 ECD bound to PTH_15-34 showing summarized Nb-binding site characterization. The structure of full-length ECD containing exon 2 (wheat) was generated from Alphafold2. This structure was aligned with the PTH_15-34 + ECD structure (PDB: 3C4M) using the "Align" command in Pymol. Source Data used to generate graphs are provided in Source Data Files.

concentrations below 10 nM (Fig. 5) and that PTH_1-11-6E did not induce arrestin recruitment at concentrations up to 10 μM (Fig. 3B). To quantitatively compare the differences observed between the two signaling pathways, the bias model of agonism was used to calculate ΔΔLog ($E_{max}$/EC_50) values (Supplementary Fig. 20). Only a crude estimate of bias factors was possible due to the weak activity of conjugates for inducing arrestin recruitment. We also found that PTH_1-11-Nb_PTHR1

failed to induce intracellular calcium mobilization (Fig. 5E, Supplementary Fig. 21), in line with findings from engineered receptors above.

These observations prompted us to test whether PTH_1-11-Nb_PTHR1-induced receptor internalization. Observation of receptor trafficking by fluorescence microscopy showed that PTH_1-11-Nb_PTHR1 was inefficient in inducing receptor redistribution into intracellular puncta relative to PTH_1-34 (Supplementary Fig. 22 and Supplementary

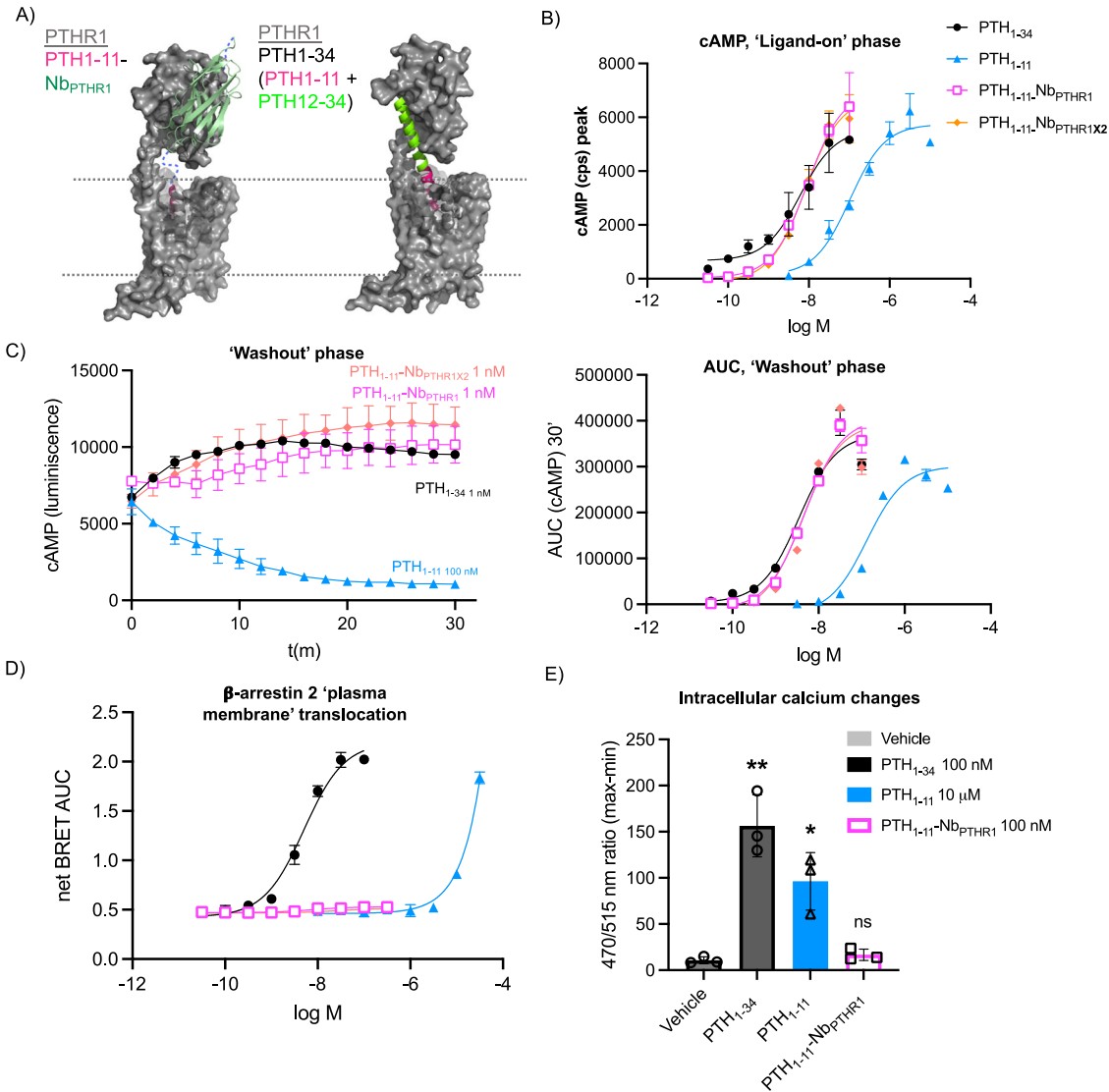

**Fig. 5 | Evaluation of Nb-ligand conjugate signaling at WT-PTHR1 (unmodified receptor). A** A schematic comparing a hypothetical mode of association between $PTH_{1-11}$-$Nb_{PTHR1}$ and PTHR1 with that of an experimental $PTH_{1-34}$-PTHR1 complex (PDB: 6FJ3). Note that this model is for schematic purposes only and is not intended to precisely depict the mode of $Nb_{PTHR1}$ binding. **B** Representative concentration-response curves for induction of cAMP responses in cells expressing PTHR1. *Y*-axes refer to maximal cAMP responses observed in a plate reader, quantified by counts per second "cps". **C** (Left) Representative kinetic evaluations of ligand-induced responses following washout. (Right) Washout signaling was quantified as AUC and characterized in concentration-response plots. For panels **B** and **C** the number of independent biological replicates is shown in Supplementary Fig. 3. **D** Representative concentration–response curves for ligand-induced recruitment of β-arrestin 2 to the plasma membrane. Endosomal β-arrestin 2 data is shown in

Supplementary Fig. 19. For panels **B–D**, data points are shown as mean ± SD from technical duplicates in a single representative experiment, fit to a three-parameter logistic sigmoidal model. Each assay was evaluated in 3–5 biological replicates with independent replicates shown in Supplementary Fig. 17. Tabulation of agonist potency and summarized AUC washout dose–response parameters are shown in Table 1. **E** Measurement of intracellular $Ca^{2+}$ mobilization in response to indicated compounds. Data points correspond to means ± SEM from three independent experiments conducted with six technical replicates per experiment. Statistical significance was assessed by one-way ANOVA, with Dunnett's post hoc correction (*$p < 0.05$; **$p < 0.01$; ***$p < 0.001$; ****$p < 0.0001$; ns not significant. Vehicle versus $PTH_{1-34}$ $p = 0.0037$ and Vehicle versus $PTH_{1-11}$ $p = 0.013$. Source Data used to generate graphs are provided in Source Data Files.

Methods). Results from microscopy experiments were corroborated by analysis with ELISA in which levels of the receptor at the cell surface were measured following exposure to $PTH_{1-11}$-$Nb_{PTHR1}$ or $PTH_{1-34}$ (Supplementary Fig. 23 and Supplementary Methods). $PTH_{1-34}$ caused a significant reduction in levels of cell surface PTHR1, whereas $PTH_{1-11}$-$Nb_{PTHR1}$ did not.

To test what role the binding of $Nb_{PTHR1}$ to receptor played in the unexpected ligand signaling properties, we added $Nb_{PTHR1}$ and peptide ligands separately (Supplementary Fig. 24), which demonstrated that the presence of $Nb_{PTHR1}$ had a negligible impact on $PTH_{1-11}$ or $PTH_{1-34}$ signaling and bias. To assess the relative contributions of Nb and

$PTH_{1-11}$ for the affinity of these conjugates for PTHR1 we compared the binding of labeled Nb with labeled $PTH_{1-11}$-Nb conjugates (Supplementary Fig. 25). $Nb_{PTHR1}$ and $Nb_{PTHR1-X2}$ stained cells with similar intensity and potency when compared to their $PTH_{1-11}$-Nb conjugates, indicating that the Nb-receptor epitope interaction provides most of the binding affinity for these conjugates. We also tested whether linking $PTH_{1-11}$ to an Nb could alter the conformational properties of either building block. Circular dichroism (CD) analysis of $PTH_{1-11}$, $Nb_{PTHR1}$, and $PTH_{1-11}$-$Nb_{PTHR1}$ conjugate demonstrated that $PTH_{1-11}$ adopts primarily a random coil confirmation, which is not dramatically changed upon conjugation with $Nb_{PTHR1}$ (Supplementary

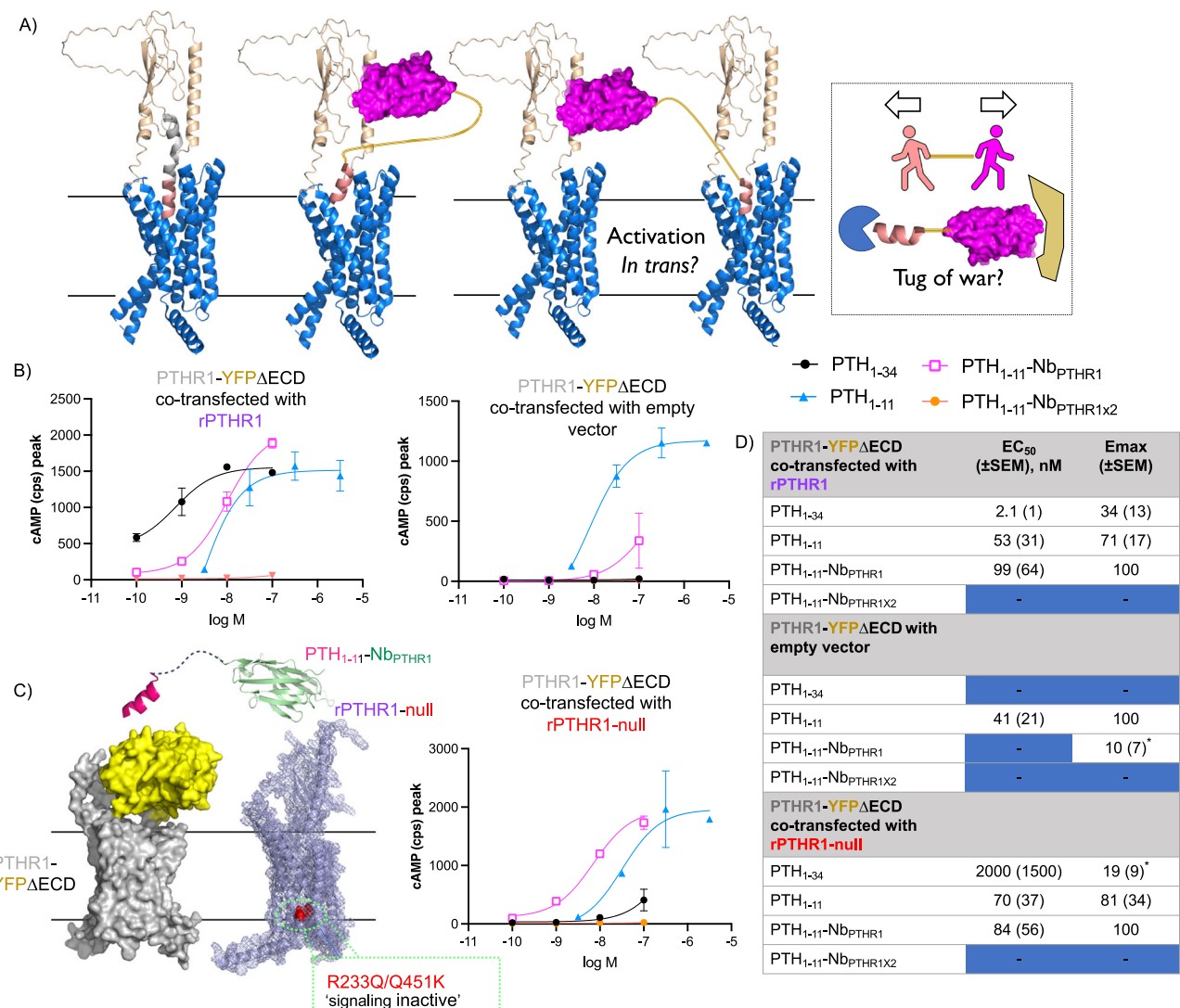

**Fig. 6 | Mechanisms of PTHR1 activation. A** Proposed mechanisms to differentiate the binding and signaling of PTH$_{1\text{-}34}$ (salmon and gray) and Nb-PTH$_{1\text{-}11}$ (salmon and purple) to PTHR1 (blue and wheat). The standard model of receptor activation consists of a single ligand activating a single receptor (left). An alternative mechanism consists of one ligand acting upon two receptors in proximity, "activation in trans" (middle). Signaling behavior might also relate to a discrepancy in linker length and the distance between binding sites for Nb$_{PTHR1}$ and PTH$_{1\text{-}11}$ in a "tug of war" type mechanism (right). **B** Representative dose–response data for induction of cAMP responses in cells co-transfected with PTHR1-YFPΔECD (PTHR1 ECD replaced with YFP) and other receptor plasmids. **C** Schematic and data for PTH$_{1\text{-}11}$-Nb$_{PTHR1}$ activation in trans in a system co-expressing PTHR1-YFPΔECD and rPTHR1-null (R233Q/Q451K rPTHR1, signaling inactive). Data in Panels **B** and **C** correspond to mean and associated SD from technical duplicates in a single representative experiment. Schematic structures of PTHR1-YFPΔECD and rPTHR1-null were generated using Alphafold2 (see the "Methods" section). Mutations in rPTHR1-null are highlighted in red. **D** Tabulation of agonist potency and $E_{max}$ parameters derived from three independent experiments shown as mean(±SEM). $E_{max}$ values were normalized to the $E_{max}$ of either PTH$_{1\text{-}11}$-Nb$_{PTHR1}$ or PTH$_{1\text{-}11}$. * indicates that the value shown corresponds to the response recorded at the highest tested concentration rather than an $E_{max}$ value generated by model fitting. Representative independent replicates are shown in Supplementary Fig. 31. Source Data used to generate graphs are provided in Source Data Files.

Fig. 26 and Supplementary Methods). The sum of CD readings recorded for PTH$_{1\text{-}11}$ and Nb$_{PTHR1}$ is similar to the CD spectrum recorded for PTH$_{1\text{-}11}$-Nb$_{PTHR1}$ conjugate, suggesting that neither building block substantially affects the conformational propensities of the partner component when conjugated.

We wondered whether the approach described above for PTHR1 could be extended to other GPCRs. To test this, we established an analogous set of reagents to study signaling in glucagon-like peptide-1 receptor (GLP1R). Nb-based tethering imparts its strongest impact when the activity of the ligand is sufficiently low such that receptor activation only occurs when the ligand is concentrated at its target via binding of the specific Nb. Like PTHR1, GLP1R is a family B1 GPCR that signals through the GαS-cAMP and β-Arr-mediated pathways and naturally binds to a large peptide ligand (glucagon-like peptide-1,

GLP1)[41]. A plasmid encoding a tagged version of GLP1R with an N-terminally engrafted 6E epitope tag (Supplementary Fig. 36 and Supplementary Methods) was used to establish a HEK293-based cell line that stably expresses GLP1R−6E and the cAMP Glosensor reporter[42]. We synthesized an analog of GLP1 with alanine mutations at two sites known to be important for high affinity binding to GLP1R ECD (GLP$_{mut}$, Supplementary Fig. 27)[43]. As expected, GLP$_{mut}$ showed reduced cAMP-induction potency compared to wild-type GLP1. Potency for cAMP signaling was augmented by conjugation with either Nb$_{6E}$ or a Nb that binds to GLP1R with high affinity (Nb$_{GLP1R}$, Supplementary Fig. 28), with these conjugates showing cAMP signaling comparable to wild-type GLP1 (Supplementary Fig. 27). With parallels to observations for PTHR1-targeting conjugates, neither Nb$_{6E}$-GLP1$_{mut}$ nor Nb$_{GLP1R}$-GLP1$_{mut}$ induced the recruitment of β-arrestin in cells

expressing GLP1R-6E (Supplementary Fig. 27). Collectively, these observations demonstrate that Nb-ligand conjugates can be readily generated that exhibit highly biased agonism.

Given the biased agonism profile of ligand conjugates for their target receptors, we sought to further study the mechanistic details of this phenomenon. One hypothesis on why PTH$_{1-11}$-Nb conjugates exhibit biased agonism is that they may engage with receptors in a manner that is qualitatively different from conventional ligands such as PTH$_{1-34}$. Although PTH$_{1-34}$ is comprised of two functional peptide domains that interact with different portions of the receptor, structural studies have shown this peptide binds to and acts upon a single receptor. We wondered whether PTH$_{1-11}$-Nb conjugates could act by bridging two separate receptor protomers ("activation in trans", Fig. 6A). In this scenario, the Nb would bind to one receptor and the linked PTH$_{1-11}$ would activate a different nearby receptor. Notably, there is evidence that receptor dimerization and oligomerization can impart biased signaling properties for class B GPCRs[44]. We sought to probe this hypothesis by extending the length of the linker between the binding and receptor activation elements of the PTH·Nb conjugates. A longer linker could help alleviate any signaling deficiency related to a "tug of war" scenario (Fig. 6A, Right) or facilitate activation in trans. Even with extended linkers, we observed little variation in signaling properties, including biased agonism (Supplementary Fig. 29). To provide a more direct assessment of activation in trans, we transfected cells to express two distinct constructs of PTHR1: an N-terminally truncated derivative of PTHR1 (YFP-delNT-PTHR1) that is bound poorly by Nb$_{PTHR1}$ but is signaling competent[27] and full-length rat PTHR1 with mutations in its transmembrane portion (rPTHR1-null, R233Q/Q451K) that renders it signaling incompetent (Supplementary Fig. 30)[45] but is still recognized by Nb$_{PTHR1}$ (Fig. 6B). PTH$_{1-11}$-Nb$_{PTHR1}$ is weakly active on cells expressing YFP-delNT-PTHR1, but upon co-transfection with rPTHR1-null its biological activity improved substantially (Fig. 6B–D, Supplementary Fig. 31). PTH$_{1-34}$, which also binds rPTHR1-null exhibited no such enhancement in activity upon co-transfection. A set of control experiments confirmed that both PTH$_{1-34}$ and PTH$_{1-11}$-Nb$_{PTHR1}$ are highly active on cells transfected with WT-rPTHR1 (Fig. 6D) and that co-expression of rPTHR1-null does not alter the expression level of YFP-delNT-PTHR1 (Supplementary Fig. 32). Experiments in an osteoblast-derived cell line (SaOS2), which endogenously express low levels of PTHR1[46], showed weak activity for PTH$_{1-11}$-Nb$_{PTHR1}$ relative to PTH$_{1-34}$ (Supplementary Fig. 33 and Supplementary Methods). We hypothesize that activation of receptors using an "in trans" mechanism would be highly dependent on receptor densities. In total, these findings offer evidence that Nb-PTH$_{1-11}$ conjugates can efficiently engage in receptor activation in trans, whereas PTH$_{1-34}$ cannot. This behavior correlates with biased agonism observed in other assays and may be mechanistically related.

## Discussion

The prevailing view on how selected ligands manifest biased agonism is through the stabilization of distinct receptor conformations (or conformational ensembles) that preferentially engage one intracellular coupling partner over another. Receptor conformational preferences are thought to be dictated by contacts engaged by the ligand binding to the orthosteric (or allosteric) site(s). In this study, we showed that a ligand of PTHR1 (PTH$_{1-11}$) that signals through all receptor-engaged pathways, albeit with moderately diminished potency for β-arrestin recruitment, can be converted to a ligand that is highly biased for signaling through the Gαs pathway through linkage with receptor binding Nbs (or peptide). Such variations in the behavior between ligand and ligand-Nb conjugates are unexpected given that both PTH$_{1-11}$ and PTH$_{1-11}$-Nb conjugates contain the same core ligand structure and differ only in whether they are linked to a receptor binding Nb. These observations expand the utility of the systematic approach for linking peptidic GPCR ligands with

antibodies (or their fragments), known as CLAMP, that was only recently described[27].

Our unexpected observation of PTH$_{1-11}$-Nb bias poses the question of how and why pathway selective signaling occurs in this context. To rule out the possibility that binding of the tethering Nb or peptide was imparting an allosteric effect on receptor function we used a variety of approaches. We used engineered receptors in which we dictated the site of Nb or peptide tethering through the introduction of an artificial binding site, either in the form of a genetically engrafted epitope tag or an engrafted nanobody. These engineered sites, placed in a non-conserved and disordered region of the receptor, are distant from the orthosteric site and any portion of PTHR1 thought to be involved in mediating its transition from an inactive to active state. Ligands conjugated to Nb or peptides that bound to these engineered sites were very highly biased in their signaling. For experiments with WT PTHR1, we characterized the receptor epitope to which two Nbs (Nb$_{PTHR1}$ and Nb$_{PTHR1X2}$) bound (Fig. 4). The PTH$_{1-11}$-Nb conjugate made from either Nb exhibited highly biased agonism at WT PTHR1, in line with results on engineered receptors. Addition of free Nb$_{PTHR1}$ (not linked to PTH) also had no impact on ligand signaling and bias (Supplementary Fig. 24). These experiments collectively suggest that biased agonism results not from Nb binding per se but rather as an emergent property from linking PTH$_{1-11}$ to an anchor that binds to the receptor at some ancillary location. This is contrasted with PTH$_{1-34}$, in which PTH$_{1-11}$ is anchored by the binding of the PTH$_{12-34}$ fragment. Note, that PTH$_{1-34}$ is more potent than PTH$_{1-11}$ in all signaling pathways measured and does not exhibit strong ligand bias. We speculate that PTH$_{1-11}$-Nb might engage different dynamics for receptor conformational changes and activation in comparison to those recently described for natural ligands of PTHR1[47]. Previous studies with splice variants of CXCR3, in which the extracellular portion is varied, demonstrate that receptor alterations that modify binding mode can lead to pathway-specific signaling outcomes[48,49].

We hypothesize that variation in ligand anchoring mechanisms contributes to the high level of signaling bias observed for the PTH$_{1-11}$-Nb conjugates. Two nonexclusive schemas can be used to conceptualize these effects. One possibility is that the topologic constraints imposed by Nb binding do not allow the simultaneous engagement of the receptor orthosteric site by the linked PTH$_{1-11}$ (Fig. 6A, "Tug of War"). This constraint might result in a transient or altered mode of binding of PTH$_{1-11}$ at the receptor orthosteric site, potentially leading to ligand bias. This hypothesis is disfavored by the observation that variation of the linker length between Nb and PTH$_{1-11}$ has little impact on the signaling properties observed (Supplementary Fig. 29). Another possibility is that Nb tethering facilitates receptor activation through a mode that involves two receptor protomers (activation in trans). We find preliminary evidence of a difference for receptor activation in trans in comparing PTH$_{1-34}$ and PTH$_{1-11}$-Nb$_{PTHR1}$, which suggests this mechanism may be important for biased agonism, consistent with reports of GPCR assemblies facilitating biased agonism[44,50]. Whether Nb-ligand conjugation provides a generalizable approach for the design of biased agonists is under study. Broad applications for targeting the GPCR superfamily will require the extension of this approach to small molecule ligands. Notably, bitopic small molecule agonists have been shown to exhibit biased agonism at mu opioid receptor[51]; however, in this case, both binding units associate with a single receptor protomer. Mounting evidence suggests that bitopic ligands that restrict GPCR conformational changes might be particularly rich sources of biased agonists[52]. Further insight into structural features of PTHR1-biased agonism is available from a recently reported structure of a Gαs-biased agonist (PCO371) bound to PTHR1[53]. It is possible that PTH$_{1-11}$-Nb$_{PTHR1}$ conjugates may elicit conformational changes similar to those seen in PCO371-bound receptors, although experimental confirmation is needed.

Efforts to apply Nb-ligand tethering to other receptors will require Nbs (or antibodies) that bind to the extracellular receptor face. At

present, there are a limited number of examples of Nbs that bind surface-exposed regions of GPCRs[54–59], and only some of these Nbs have been structurally characterized[60–62]. New Nb screening technologies[63,64] are emerging and will likely facilitate further progress. Another exciting possibility is that receptor activation in trans could operate across GPCR heteromers or even assemblies incorporating plasma membrane-localized proteins that are not GPCRs. Caution is needed as past work has shown that activation in trans is not possible for all GPCR-membrane protein pairs expressed on the same cell surface[65]. Even with these caveats, the prospects of using straightforward conjugation methodology to generate highly active, highly biased ligands offer compelling prospects for future studies.

The development of compounds that target GPCRs and induce biased signaling responses is of substantial interest for therapeutic development. Despite this interest, the paucity of pathway-selective ligands among approved therapeutics highlights uncertainty about translational prospects. Biased ligands may serve to induce preferential activation of receptor-coupled pathways that promote therapeutic effects while minimizing signaling through pathways that mediate drug-related side effects[6]. Efforts to explore such applications are limited when suitably biased agonists are unavailable. Alternative methods, such as genetic knockout of GPCR signaling partners, offer insights into the consequences of biased agonism. For PTHR1, the deletion of β-arrestin 2 in mice prevented bone loss in response to continuous PTH stimulation, a side effect observed with conventional PTH-based therapeutics. This observation suggests that PTHR1 agonists biased towards Gαs might be candidates for osteoporosis therapeutics with reduced side effects. In this regard, it is noteworthy that the PTH$_{1-11}$-Nb conjugates reported here appear to be the most highly Gαs-biased PTH agonist peptides reported to date. It is also worth noting that nanobody conjugation may be a general strategy to consider for delivering otherwise weak or unstable potential drug products to their intended target sites of action. Future efforts to generate biased agonists and related therapeutic lead candidates might benefit from the approach and principles outlined here.

## Methods

### Mass spectrometry
Mass spectrometry data was acquired on a Waters Xevo qTOF LC/MS instrument. Certain samples were subjected to reverse-phase LC (Hamilton PRP-h5 column, 5 µM particle size, 300 Å pore size). All samples were analyzed in positive ion mode. Peptide and protein identity was confirmed by mass spectrometry upon each new preparation. Samples were analyzed in positive ion mode. For proteins analyzed by mass spectrometry, singly charged ions were not observed, so protein intact mass was calculated from the analysis of multiply charged ions using the MaxENT algorithm on MassLynx (Waters) software. Mass spectra are shown in Supplementary Data 1.

### Nanobody expression and purification
Nb protein sequences acquired from literature (previously named 22A3 and 23A3)[36] were codon optimized for bacterial expression and cloned into a pET26b expression in frame with pelB and His6 sequences using clone EZ service from GenScript. The production and purification of Nb$_{6E}$ (previously named VHH05) has been described previously[27]. Briefly, BL21(DE3) E. coli were transfected via heat shock with plasmids encoding nanobodies of interest and grown in a medium (Terrific Broth) containing kanamycin (50 µg/mL). Transformed bacteria were used to generate a starter culture, which was used to inoculate full-size cultures (1–4 L) containing kanamycin. This culture was grown at 37 °C, and expression was induced with Isopropyl β-D-1-thiogalactopyranoside (IPTG, 1 mM) at an optical density (OD$_{600}$) of 0.6. The induced culture was then shaken at 30 °C overnight.

Bacteria were harvested via centrifugation for 30 min at 6000 RPM (9225 × g) (Avanti J Series centrifuge) and resuspended in 30 mL of NTA wash buffer (tris buffered saline + 10 mM imidazole, pH 7.5) containing protease inhibitor (Pierce Protease Inhibitor Tablets, Thermo Fisher A32953). Cells were then lysed via sonication, and the lysate was centrifuged at 15,000 RPM (32,000 × g) for 45 min. After centrifugation, the Nb was purified from the lysate by batch-based Ni-NTA chromatography, followed by size-exclusion chromatography (Cytiva Akta/ Pure) using a HiLoad 16/600 Superdex 200 pg column with an isocratic gradient of TBS (flow rate 1 mL/min). Fractions of interest were collected and concentrated using a 10 kDa MW cutoff Amicon spin-concentrator. The identity of the purified fraction was confirmed by mass spectrometry, and the concentration of Nb was determined by measuring the absorbance at 280 nm. Aligned sequences of Nbs used in this study are shown in Supplementary Fig. 34. Information for expression and purification of GLP1R extracellular domain is described in Supplementary Methods and Supplementary Fig. 15.

### Peptide synthesis
All peptides, unless specified otherwise, were synthesized via solid phase peptide synthesis with Fmoc protection of the amine backbone on a Gyros PurePep Chorus or Liberty Blue Microwave-Assisted Automated Peptide Synthesizer. The GLP1 and Exendin-4 peptide were purchased from Abcam (#ab142024 and # ab120214). Where relevant, a Cys residue was incorporated at the C-terminus of the peptide for functionalization using Cys-maleimide chemistry (see below). Peptide synthesis was performed on Rink Amide resin (0.05 mmol scale) to afford a C-terminal carboxamide. Fmoc-amino acids were dissolved in dimethylformamide (DMF) and added to resin (8 equivalents) with HATU ((1-[Bis(dimethylamino)methylene]-1H-1,2,3-triazolo[4,5-b]pyridinium 3-oxid hexafluorophosphate, 8 equivalents) and N,N-diisopropylethylamine (DIPEA, 16 equivalents). Fmoc groups were deprotected using 20% piperidine in DMF.

Upon completion of the synthesis, peptides were cleaved from the resin using a cleavage cocktail comprised of trifluoroacetic acid(-TFA)/H$_2$O/triisopropylsilane(TIS) (92.5:5:2.5% by volume) and rocked at room temperature for 3 h prior to filtration. After the cleavage, the crude peptides were precipitated using chilled diethyl ether and pelleted by centrifugation (3000 × g for 2 min). Peptides were purified via preparative-scale HPLC using a Phenomenex Aeris Peptide XB-C18 Prep column (particle size 5 µM, 100 Å pore size) with a linear gradient of solvent A (0.1% TFA in H$_2$O) and solvent B (0.1% TFA in acetonitrile). Fractions containing peptides of interest were identified using mass spectrometry analysis. Mass spectrometry characterization of peptides is shown in Supplementary Table 1. Fractions of interest were combined and lyophilized. Lyophilized peptides are then dissolved in DMSO at desired concentrations and frozen. Sequences for selected peptides used in competition binding assays are shown in Supplementary Fig. 35. The peptide sequences used in functional assays are different from those used in the competition binding assays.

Peptides with C terminal Cys residues purified by HPLC were incubated with 3 molar equivalents of DBCO-maleimide (Click Chemistry Tools #A108-100), and the DBCO-peptide products purified by HPLC. The purified product was lyophilized and dissolved in DMSO at a concentration of 1 mM. The identity of DBCO-modified PTH peptides was confirmed by LC−MS.

### Nanobody labeling via sortagging
Sortagging reactions were performed as previously described[27] and were comprised of the following components: protein bearing a sortase recognition motif (LPETGG) followed by a hexa histidine tag at the C-terminus (20–200 µM final concentration), triglycine-probe conjugates (500–1000 µM final concentration), and Sortase 5 M (10–20 µM final concentration). Reactions were performed in sortase buffer (10 mM CaCl$_2$, 50 mM Tris, 150 mM NaCl, pH 7.5) and shaken at 12 °C overnight. After incubation, the reaction was incubated with nickel

NTA beads to capture Sortase 5 M and unreacted starting protein. Uncaptured material was further purified using disposable desalting columns to remove triglycine conjugates (Cytiva PD-10 Sephadex™ G-25M). Eluents were monitored for absorbance at the fluorescent probe absorbance wavelength (tetramethylrhodamine: 555 nm, fluorescein: 494 nm), 220 and 280 nm for the presence of protein conjugate. Fractions containing the product were combined and then concentrated by spin filtration (Amicon Ultra 0.5 mL centrifugal filters 10,000 NMWL).

## Click chemistry preparation of Nb-ligand conjugates

The peptide-DBCO conjugate was used in an azide-alkyne ("click") reaction between the azide-functionalized Nb and a DBCO-modified synthetic PTH peptide (Fig. 1), as previously described[27]. Nb-biotin-azide conjugates were mixed with an excess of PTH-DBCO (3-fold molar excess) in TBS. The reaction was shaken at 25 °C until unreacted Nb-biotin-azide had been completely consumed. The product conjugates were purified from free DBCO-modified peptide using a PD10 size exclusion column. Product identity was confirmed by LC–MS.

## Cell culture and receptor constructs

A Human Embryonic Kidney 293 cell line (HEK 293; ATCC CRL-1573) stably transfected with luciferase-based pGlosensor-22F cAMP reporter plasmid (Glosensor, Promega Corp.) has been described previously [GS22[66]]. GS22 cells were used to generate cell lines stably expressing either native human PTHR1, PTHR1-6E, PTHR1-Nb$_{6E}$, or GLP1-6E as previously described[27,29]. Plasmids encoding receptors under study have been previously defined[27]. Sequences of all plasmids validated by Sanger sequencing. All cell lines were cultured in high-glucose DMEM (Gibco, Thermo Fisher Scientific), containing 10% fetal bovine serum (Sigma-Aldrich) and 1% of penicillin and streptomycin mixture and grown in an incubator at 37 °C in a humidified atmosphere containing 5% $CO_2$. The cells were passaged every 3–4 days and seeded to achieve a confluency of 60–70% for experiments relying on transient transfection. Cells plated for luciferase-based cAMP assays were grown in the same condition but were seeded to full confluency prior to assay execution. All cell lines were regularly checked for mycoplasma infection using the Lonza MycoAlert mycoplasma detection kit and were found to be negative. Aligned sequences for rat versus human PTHR1 are shown in Supplementary Fig. 36.

## Luminescence-based live cell cAMP accumulation assay

Cellular cAMP production was measured in living cells as described previously. In brief, cells in culture were trypsinized and transferred into clear bottom white-walled 96-well plates at a density of 80,000 cells per well. After achieving confluency, the culture medium was removed, and a $CO_2$-independent medium containing luciferin (0.5 mM) was added. Luminescence was measured until a stable background reading was obtained (~10 min). Serial dilutions of ligands were then added such that the final well volume was 100 μL, with luminescence measured in real-time every 2 min for 12 min (Biotek Neo 2 plate reader). The peak luminescence responses (typically measured at 12 min) were used to generate concentration-response curves. The concentration–response curves were fitted to individual experiments utilizing a sigmoidal dose-response model (log[agonist] vs. response [three parameters]; GraphPad Prism) to produce $EC_{50}$ values. In instances where curves did not reach the plateau at the highest concentration tested, curves were constrained to the $E_{max}$ value observed for an index ligand run simultaneously.

The signaling duration of ligands was evaluated using "washout" assays in which the cells were stimulated with an agonist for a defined period as described above (ligand-on phase). After this period, the medium containing ligands was discarded. A new $CO_2$-independent medium containing fresh luciferin was added to all wells (resulting in a 2–5-min interlude between ligand on and washout periods), and luminescence responses were recorded for an additional 30 min (ligand-off phase). For competition-washout assays, the experimental protocol was as described above, except that competitors were introduced during the ligand-off phase. Area under the curve (AUC) values, measured using GraphPad Prism, were used to construct dose–response curves for the washout assays.

For the dimerization luminescence-based cyclic AMP assays, HEK293 cells expressing a variant of PTHR1 with the extracellular domain replaced with yellow fluorescent protein (YFP)[27] were transfected with 1 μg of either HA-tagged wild-type rat PTHR1[45], R233Q/Q451K rat PTHR1[37], or no receptor. After transfection, the cells were grown and assayed for cyclic AMP production, as described above.

## Flow cytometry analysis for Nb binding to receptors expressed in HEK293 cells

HEK293 cells stably expressing either native human PTHR1, PTHR1-6E, or PTHR1-Nb$_{6E}$ were cultured as described above. Cells were harvested by trypsinization, which was quenched with the addition of DMEM/FBS. Subsequently, the cells were transferred to a round bottom 96-well plate, pelleted by centrifugation (500 × g for 3 min), and resuspended in PBS containing 2% BSA (w/v) (PBS/BSA). Cells were incubated with varying concentrations of Nbs labeled with biotin using sortagging. Following incubation on ice for 30 min, cells were pelleted, washed, and resuspended in PBS/BSA containing streptavidin-APC (1:2000 dilution in PBS/BSA, (BioLegend #405207)) and incubated for 30 min on ice prior to washing. Washed cells were then resuspended in PBS/BSA for analysis by flow cytometry on a CytoFlex flow cytometer (Beckman Coulter). Samples were analyzed using FlowJo (version 7.6). Intact cells were identified based on their forward scatter/side scatter profile (Supplementary Fig. 37), and staining intensity was monitored in the APC channel (FL4). A minimum of 2000 events corresponding to intact cells were recorded. Flow cytometry histograms were used to calculate median fluorescence intensity (MFI) values in each sample. MFI values were averaged among replicate samples.

## Bioluminescence resonance energy transfer (BRET) assays

**Effector membrane translocation assay to measure β-arrestin and Gαs trafficking.** HEK cell lines expressing receptors of interest were passaged and transfected as described above, with specific modifications based on a previously described protocol[34]. One day after seeding cells in a 10 cm dish, a transfection cocktail containing membrane-tethered BRET acceptor plasmid (rGFP-CAAX or rGFP-FYVE, 1008 ng) and RlucII-conjugated β-arrestin 2 BRET donor plasmid (β-arrestin 2-RlucII, 72 ng) and Lipofectamine 3000 was added following manufacturer instructions. For G-protein dissociation BRET assays, 1008 ng of rGFP-CAAX plasmid and 144 ng of Gαs-RlucII plasmid were used. Transfected cells were seeded at a density of 60,000 cells per well in a white 96-well plate and grown overnight. Prior to assays, the culture medium was removed and replaced with Hanks buffered saline solution (HBSS) supplemented with 5 mM HEPES. Cells were then treated with varying concentrations of ligand mixed with the luciferase substrate coelenterazine prolume purple (1 μM, NanoLight Technologies). The ligand concentration-dependent BRET change was measured using a Biotek Neo 2 plate reader. Signal at donor and acceptor wavelengths (410 and 515 nm, respectively) were measured every 150 s for a total of 30 min. BRET ratios (515/410 nm) were calculated and analyzed in GraphPad Prism v9. AUC values from kinetic measurements were used to generate concentration–response plots.

**Measurement of ligand binding kinetics using BRET.** A cell line stably expressing a construct encoding nanoluciferase fused to the N-terminus of the PTHR1 was used as previously described[38]. This cell line was plated in black-walled 96-well plates, as described above. On the day of the experiment, the cell culture medium was removed, and

HBSS supplemented with 0.02% $NaN_3$ and 5 mM HEPES was added (100 μl per well). The cells were then incubated in this buffer for at least 30 min at room temperature prior to the experiment. TMR-labeled peptides or Nbs were prepared at varying concentrations in HBSS/HEPES buffer containing 5 μM coelenterazine-h (NanoLight Technologies). Azide-containing medium was removed from the assay plate, and 100 μL of the peptide solution with coelenterazine-h was added to each well. Luminescence measurements were taken at 450 nm and 610 nm at 12 s intervals for 5 min (except for kinetic binding assays, described below).

In the washout (competition) assay mode, TMR-labeled Nb or peptide was first added at varying concentrations, followed by the addition of unlabeled ligands. Signals were recorded in two separate intervals: once when the labeled ligand was added alone and once again after the unlabeled competitor was added. AUC of kinetic curves was determined to quantify changes in binding caused by competitor addition. Alternatively, in the pre-incubation competition assay format, varying concentrations of unlabeled ligands were added simultaneously with TMR-labeled Nb or peptide. In this assay format, the impact of competitors on labeled ligand binding was quantified by measuring the BRET signal 12 min after addition.

For kinetic binding assays, BRET readings (at 450/610 nm) were taken in 7 s intervals for a total of 5 min following ligand addition. To measure the kinetics of ligand binding, the time course graphs were analyzed via the one-phase association method in GraphPad Prism 9, from which the $k$(obs) values were obtained (Eq. (1)) and plotted as a function of concentration onto a linear regression graph, which was used to derive the $k$(on) and $k$(off) values (Eq. (2)), which were, in turn, used to derive the $k$(D) value (Eq. (3)), as previously described[38].

$$Y = Y_0 * (1 - e^{k(obs)t}) \tag{1}$$

$$k_{obs} = k_{on}[L] + k_{off} \tag{2}$$

$$k_D = k_{off}/k_{on} \tag{3}$$

### Ligand-induced intracellular calcium mobilization assays

The activity of various ligands for inducing intracellular calcium mobilization $[Ca^{2+}]_i$ was measured using Calbryte 520 AM (AAT Bioquest, USA) using the stably transfected cell lines described above. Briefly, 24 h prior to the assay, cells were seeded into 96-well black-walled plates (Thermo Fisher Scientific) and incubated overnight at 37 °C. Calbryte 520 AM dye was dissolved in an HBSS in the presence of 0.04% Pluronic F-127, 20 mM HEPES, and 2.5 mM probenecid, which was added to each well and incubated at 37 °C for at least 1 h before the assay. Dye loading solution was then removed and replaced with HBSS. Fluorescence was measured every 2 seconds ($\lambda_{excitation}$ = 470 nm, $\lambda_{emission}$ = 515 nm) using a high-throughput FLIPR$^{TETRA}$ cellular screening system (Molecular Devices). Peptides were added after at least 2 min of background recording. Responses were quantified as relative fluorescence ("max−min") calculated in ScreenWorks software (Molecular Devices) and normalized to signal background.

### Hydroxy radical footprinting and analysis of nanobody–PTHR1 ECD interaction

Protein samples for analyzing Nb−PTHR1 ECD interactions were prepared, treated, and analyzed as previously described[39] with some modifications. PTHR1 extracellular domain (ECD, see Supplementary Fig. 15 for sequence) was produced and purified as previously described[20]. Nb$_{PTHR1}$ or Nb$_{6E}$ were incubated with PTHR1 ECD (1:1 molar ratio, 5.4 μM) at room temperature for 30 min prior to labeling. Samples were exposed to plasma-induced modification of biomolecule (PLIMB) treatment conditions for 20 s. Following labeling, samples were quenched with a 5 μL solution of 250 mM methionine in PBS

(pH 7.4). Following PLIMB exposure, samples were proteolytically digested into peptides with trypsin. Samples were subjected to solid phase extraction using C18 StageTips and then analyzed using data-dependent acquisition with an Orbitrap Exploris 240 mass spectrometer.

The '.raw' mass spectrometry data files were searched against the PTHR1 ECD sequence using the Protein Metrics Oxidative Footprinting Module. A list of standard expected modifications and expected PLIMB modifications was utilized in the database search. Peptides were identified using MS and MS/MS spectra, setting a 1% false discovery rate (FDR) cutoff. Changes in solvent accessibility for the trypsin-digested samples were determined via comparison of the sum normalized intensities.

### Alphafold enabled receptor structural modeling

Structural models of the engineered receptors were generated through input protein sequences into an Alphafold2 Colab notebook[67]. The default AlphaFold2 settings were applied, which generates five models for each input sequence. Requisite models were applied to generate all graphics. Models were downloaded in PDB format and prepared as graphics using Pymol. In some cases, models were aligned with related experimentally determined structures using the "align" command in PyMOL.

### Data and statistical analysis

Data presented are either representative data from a single experiment (performed in technical duplicates, expressed as mean ± SD) or averaged (combined) data from at least three biological (independent) replicates (expressed as mean ± SEM). These distinctions are described in figure captions. Statistical analyses were performed only on collated data from biological replicates with $n \geq 3$ using GraphPad Prism v9. Ligand bias (Fig. 4) was quantified by calculating the intrinsic relative activities of various ligands compared to a reference agonist (PTH$_{1-34}$) using Eq. (4) shown below[68]:

$$\text{Bias factor} = \log\left(\left(\frac{E_{MAX.1}}{EC_{50.1}}\frac{EC_{50.2}}{E_{max.2}}\right)_{lig}\left(\frac{E_{MAX.2}}{EC_{50.2}}\frac{EC_{50.1}}{E_{MAX.1}}\right)_{ref}\right) \tag{4}$$

Parameters input into this equation were derived from curves fitted to graphs comprised of data from all biological replicates. In cases where ligands demonstrated no detectable response in specified assays, an estimated value for EC$_{50}$ was generated from the dose-response curve described above.

Statistical significance ($P < 0.05$) was assessed using Student's t-test (two-tailed), or one-way ANOVA as indicated in specific figure legends.

### Reporting summary

Further information on research design is available in the Nature Portfolio Reporting Summary linked to this article.

## Data availability

All data generated in this study are provided in the Supplementary Information, Supplementary Data, and Source Data files. Previously published structural models used in this work are available from the Protein Data Bank under accession numbers 6FJ3 and 3C4M. Source data are provided with this paper.

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

## Acknowledgements

We acknowledge the Massachusetts General Hospital peptide synthesis core facility (A. Khatri) for the production of selected peptides. We thank M. Bouvier (University of Montreal) for the provision of plasmids used for BRET-based assays of β-Arrestin and G-protein signaling. We thank T. Dean (Massachusetts General Hospital) for technical assistance with microscopy experiments. We acknowledge S. Gellman (University of Wisconsin-Madison) for the provision of a construct encoding PTHR1 ECD and a cell line stably expressing nLuc PTHR1. We acknowledge the intramural mass spectrometry core facility in NIDDK (J. Lloyd) for the characterization of peptides and conjugates. The schematic in some figures was generated using Biorender.com. This work was supported by the NIH Intramural Research Program (NIDDK, 1ZIADK075157, R.W.C.), PO1-DK11794 (T.J.G.), and funding from the NIH Director's Innovation Challenge Award.

## Author contributions

The paper was written by S.S., T.J.G., and R.W.C. Conceptual input was provided by S.S., T.G., and R.W.C. Experiments were designed and performed by S.S., B.C., T.G., and R.W.C.

## Funding

## Competing interests

R.W.C. and T.J.G. are inventors on a patent (US11878063B2, granted) filed by Children's Medical Center Corporation related to the use of nanobody-ligand conjugates for biomedical applications. The other authors declare no competing interests.
