## [Peer Review File · Nature Communications]

Highly biased agonism for GPCR ligands via nanobody tetheringREVIEWER COMMENTS

Reviewer #1 (Remarks to the Author):

This is an interesting manuscript that describes the development of nanobody-tethered agonists at the PTH1R that display strong Gs bias (with a loss of beta-arrestin and Gq activity (as assessed by calcium)). The authors start by characterizing a nanobody that recognizes the 6E motif and then show the same behavior is true of PTH1-11 linked to nanobodies that target the ECD of the native PTH1R. The binding of those two nanobodies was also evaluated in detail with approaches that included hydroxy radical-based footprinting (not very helpful) and other approaches. There are a considerable number of experiments that are primarily of a technical nature focused on the mode of binding of the agonist, most interesting of which is the experiment which shows that some of this binding can occur in trans across a receptor dimer (although it does not exclude that the primary mode of activity is through a receptor monomer).

I think the studies have been performed well and are rigorous. My main question is regarding the mode of signaling of these conjugates, as the Vilardaga group has shown in a series of manuscripts that prolonged Gs signaling is usually associated with signaling from internalized receptors. This is contrast to this study, where what appears to be prolonged signaling (at least from the kinetic tracings in Fig. 2B) is likely from the plasma membrane due to the lack of barrestin activity and receptor internalization. I think this is an important question that needs to be considered as it would have a significant impact on the predicted biological activity of these compounds, as the Vilardaga group has shown that internalized Gs signaling is what is required for the bone growth response.

Another question is regarding the observed loss of Gq signaling – this was assessing by calcium. Is this also true of the conjugates to the native receptor?

Lastly, this behavior recalls something seen at chemokine receptors. Chemokines bind use a similar model of binding with an N-terminal binding to an orthosteric site and the globular domain binding the ECD. At CXCR3, there is a splice variant (CXCR3B) of the extracellular domain which results in a longer N-terminus. This change in binding in the ECD results in a receptor that is completely b-arrestin-biased (this has been shown by the Sakmar and Rajagopal labs). Here, it looks like changing the interaction in the ECD results in a Gs-protein-biased agonist.

Other comments:

Line 125-126: Should Nb6E-PTH1-11 be PTH1-11-Nb6E?

Table 1: What are the units? I assume nM.

Figure 6B: The color scheme could be improved to help differentiate between PTH1-11 and PTH1-11-NbPTHR1

Reviewer #2 (Remarks to the Author):

The manuscript entitled “Highly biased agonism for GPCR ligands via nanobody tethering” by Sachdev et al. describes the generation and characterisation of synthetic fusions of nanobodies with the n-terminus of the parathyroid hormone (peptide, PTH) (plus a control where the nanobody is inserted into the parathyroid hormone receptor (PTH1R) and the nanobody epitope is instead fused to PTH n-terminus). These chemical biology derived ligands are then characterized for a subset of their activities in recombinant cells using a variety of different PTH1R constructs.

I have substantial reservations regarding the claims of novelty, data presentation and interpretation, which I detail below. None of the primary contentions of the paper are supported by the data and the argument that the approach provides a facile means to generate biased ligands is not generally true. Many pharmacologically relevant GPCRs have small molecule agonists and little or no extracellular domain, thus an approach using an extracellular directed nanobody (which needs to be developed for every single receptor) coupled to a peptide (only a subset of receptors) does not really fulfill the description of facile or generalizable. There is also no data presented that supports the contention that these ligands will act in a biased manner on an endogenously expressed PTHR1 in a native cell. Thus, even if the data were indisputable, they would be of purely phenomenological value.

The concept of signal bias at GPCRs has been very popular for at least 25 years, yet despite this there is little evidence that supports physiologically relevant coupling of GPCRs to more than one primary transducer. The failure of Olceridine to provide a substantial improvement in safety window over other opioids should be instructive in the caution with which the GPCR field should have with this concept.

The authors claim that their approach, to use sortase mediated ligation of a nanobody with a synthetic peptide, is novel technology (they call this “CLAMP”). This claim is undermined by the fact that sortase mediated protein-protein (and protein-peptide) ligation has been described for at least 19 years: Mao H, Hart SA, Schink A, Pollok BA. Sortase-mediated protein ligation: a new method for protein engineering. *J Am Chem Soc.* 2004 Mar 10;126(9):2670-1. doi: 10.1021/ja039915e. PMID: 14995162. And is further undermined by the fact that the authors have previously described conjugation of PTH peptides to nanobodies in this journal: Cheloha RW, Fischer FA, Woodham AW, Daley E, Suminski N, Gardella TJ, Ploegh HL. Improved GPCR ligands from nanobody tethering. *Nat Commun.* 2020 Apr 29;11(1):2087. doi: 10.1038/s41467-020-15884-8. PMID: 32350260; PMCID: PMC7190724.

In figure 2 the authors provide data for cAMP responses. This data needs a control of un-modified PTH1R to allow the reader to compare the response of PTH1-34 and 1-11 at this receptor compared to the modified receptors reported. Without this control the data is very difficult to interpret. The y-axis and figure legend need to be fixed for panels A and D, is this AUC or peak? If AUC this data is inconsistent with panels C and F, where the magnitude of the response is between 5 and 12 times higher after “washout”. The data from the “washout” experiments is almost uninterpretable, the EC50 for PTH1-11-Nb6E at PTHR1-6E is not reported, but appears to be in the order of 10 nM (this is reported as being in table 1, which is actually mass-spectrometry data). This ligand appears to be a full agonist, as assessed by panel A, which means one would expect the pEC50 to be to the left of the apparent affinity, since the output being measured is amplified downstream. The affinity of this ligand would therefore be predicted to result in relatively rapid unbinding from the receptor (compare say with figure 4C). It is unclear from the methods section exactly what the protocol is for this “washout” experiment, but it seems that the idea is infinite dilution, in which case the subsidence of signal should reflect ligand unbinding. Since the authors are claiming bias away from beta-arrestin (and therefore, presumable internalisation) this sustained response is not being attributed to a pool of internalised receptors but rather persistence in ligand binding. Minimally the authors need to binding assays to determine the association and dissociation rates of these ligands as well as PTH1-34 at these engineered receptors. The “washout” data in panel E is inconsistent with cAMP data shown above in panel D; in panel D, 1 nM of PTH1-11-6E is saturating for signal, 1 nM PTH1-34 is approximately an EC50 (and less than 50% of PTH1-11-6E max) and PTH11 is a approximate EC80, yet in panel E all 3 of these concentrations start at the same point. Accompanying this figure is Supp Figure 1, where the authors appear to report that in a “washout” (infinite dilution??) experiment a competitive inhibitor causes the cAMP signal to decrease more rapidly. This is inconsistent with the general model for binding via mass action, where a competitive antagonist would NOT be expected to change the unbinding rate of an agonist. This needs to be explained. The authors should not present representative data from 1 experiment, they must present data using the mean values of the technical replicates from separate independent experiments. In the legend the curve-fit is reported as being from a 3-parameter logistic equation, this is clearly not true, the hill slope is clearly floated for PTH1-11 (panels A & D) and PTH1-11-6E (panel D) and may be floated for other panels. At present this data is not suitable for publication.

Figure 3. Again, these assays need a control of un-modified PTHR1 to allow assessment of data. The data in panel A is inconsistent with cAMP data presented in figure 1. As the authors correctly identify, cAMP is an amplified downstream signalling measure. The potency for cAMP should be (minimally) equivalent to that for Galphas recruitment but more typically to the left. This is not the case, and the authors do not address this apparent inconsistency. Calcium data needs to minimally be reported as concentration-response curves and preferably representative kinetic traces should be shown (at least in supplementary). Calcium mobilisation is typically downstream of Galphaq, but is known also to be downstream of other transducers, without the use of a specific Galphaq inhibitor, this data could have alternative interpretations. Data is again shown as mean and error of technical replicates, this is not appropriate. While the use of *, ** etc. to indicate different thresholds of statistical significance is common practice, however it demonstrates a misunderstanding of statistics and should be strongly discouraged.

Figure 4. Panel A requires specificity controls, all ligands need to be tested against cells expressing an irrelevant receptor (e.g. another Class B GPCR). For ligand binding assays data needs to be provided that demonstrates the equivalent function of nLuc labelled PTH1R with unlabelled receptor and all assays require a specificity control of either a nLuc labelled irrelevant receptor OR an excess of unlabelled ligand to establish un-specific binding. Panel B is data from a binding assay using the nLuc tagged PTH1R, the figure legend needs to be rewritten to make this clear. Since this is a saturation binding assay there needs to be a specificity control (above). It is also more usual to display this type of data on a linear scale with a hyperbolic curve and to fit to a saturation binding equation (taking into account unspecific binding). In the text, figure 5B is referred to as cAMP data (line 274, page 10), which clearly it is not and I do not see cAMP data for these constructs elsewhere in the manuscript. Panel C needs specificity control. The 2 μ M PTH1-34 addition makes two issues very evident; firstly there is some unspecific binding of the probe, and second the unbinding rate of the NbPTHR1-TMR ligand is extremely fast (consistent with the estimated low affinity of \sim 500 nM and the fast Kobs shown in supp fig 7.) – this is extremely difficult to reconcile with the “washout” data shown elsewhere in the paper – particularly figure 5, where the ligands would be expected to almost immediately unbind from the receptor. The competition experiments shown in the bar-graph (no label but next to panel C) have no indication either in the method nor figure legend of the concentration of competitor, the reader has to go to supp fig 9 to find concentrations, please add concentrations to figure legend. Supp fig 9 data is very problematic; 3 concentrations of NdPTHR1-TMR ligand are used in each competition experiment with a high degree of variability in both window and apparent affinity. For example the maximum signal in panel C is \sim 0.028, whereas this is 0.06 in panel F (Supp fig 9). More concerning is that the apparent probe affinity varies between experiments, panel C appears to have at least a 10 fold, but more likely a 30 fold higher affinity for the probe ligand compared with panel E (Supp fig 9). In D, why were more concentrations of unlabelled competitor included and the data fitted to a competition binding model to enable estimation of affinity?

Figure 5. “Washout” phase data is inconsistent with binding data, this needs to be addressed (see above). Calcium data needs to be presented as concentration-response. Representative data needs to be removed and complete, pooled data set from means of individual experiments needs to be reported.

Figure 6. Why is the magnitude of the response so much lower than all other assays (less than 1/6th by my estimation)? There is not an appropriate control for this assay set. Since the responses are so much lower than all other reported data it is feasible that the rPTHR1-null is capable of signalling at substantially reduced levels, which would mean all the signalling being measured is simply happening in-cis rather than in-trans. A control where this rPTHR1-null receptor is tested alone is required for interpretation of the data. Representative graphs need to be removed and complete, pooled data set from means of individual experiments need to be reported.

Minor comments:

Please provide the LC-MS traces for all compounds so that the reader can assess purity. (supplementary tables 2 and 2).

Please provide supplementary data using a cell-permeable cAMP analogue (e.g. 8-CPT) to ensure the GlowSensor cAMP biosensor is not saturated at maximal PTHR1 ligand concentrations.

EC50 data is shown with linear SEM, presumably the data is log-normally distributed?

Reviewer #3 (Remarks to the Author):

The manuscript by Sachdev et al (Highly biased agonism for GPCR ligands 1 via nanobody tethering) presents interesting pharmacological observations around linking the "activating" N-term sequence of PTH peptide to PTHR1 ectodomain-binding nanobodies (Nbs). In full length PTH peptide, it is well established that the C-term peptide sequence binds to the PTHR1 ectodomain with high affinity, thus allowing the weaker affinity N-term sequence to engage with the PTHR1 transmembrane domains and activate the receptor. Sachdev et al replaced the C-term PTH sequence with a range of Nbs that could either bind to engineered epitope tags in PTHR1, or natural epitopes in PTHR1. In each case, Nb fusion resulted in improved potency of the activating peptide at PTHR1 and a perceived loss of beta-arrestin efficacy compared to the activating peptide alone. These data shed light on the mechanisms by which PTH binds and activates PTHR1, and perhaps a means to engineering interesting agonists at Class B GPCRs, and thus is worthy of publication. The authors conclusions on mechanism, however, are partially flawed and should be modified by addressing the concerns below.

Kobayashi et al (doi.org/10.1016/j.molcel.2022.07.003) identified that PTH retains significant dynamics when bound to PTHR, and that these dynamics (especially partial activating peptide sequence dissociation) play a key role in G protein activation. The authors should discuss the relevance of their data to Kobayashi's findings, since Nb fusion would certainly change the dynamics of PTH1-11 when bound to PTHR1.

What is known about the folding of PTH1-11 and the influence of Nb fusion on alpha helix formation? It is becoming apparent for similar peptide receptors that peptide folding is influenced by receptor binding, and that peptide folding dynamics are important for receptor efficacy and signaling bias. The authors could perform CD spectroscopy or NMR to reveal any influence of Nb fusion on PTH1-11 folding.

The complete lack of efficacy of PTH1-11-Nb_{neg} is a concern. Shouldn't this fusion have similar efficacy to PTH1-11 alone? Have the authors tried a range of non-binding Nbs? Again, analysing the influence of Nb fusion on PTH1-11 fold may help reveal what happened to this control.

PTH1-11 seems to be less efficacious on PTHR1-6E compared to PTHR1-Nb6E, especially at beta-arrestin. Any ideas why? The authors also need to test PTH1-11Nb6E and PTH1-11-Nb_{neg} at wtPTHR1 at all pathways to be able to properly comment on their findings (i.e. are they also biased at the wt receptor?).

The insertion point of the 6E epitope was proposed to be unstructured in the receptor, yet the authors identified that NbPTHRX2 likely binds to this site. Does the fact that a Nb was identified that binds to this region suggest that it may have indeed have fold?

Fig 5D, PTH1-11-NbPTHR1 concentration was too low to see whether this molecule could act at Beta-arr, why? If the authors could have gone as high as PTH1-11, what would they expect?

- P9L279 "Analogously to experiments with engineered receptors, both NbPTHR1- and NbPTHR1-X2-PTH1-11 conjugates displayed negligible recruitment of β -arrestin 2 to the plasma membrane or early endosomes in cells stably expressing hPTHR1 (Fig. 5D-E, Supporting Figure 12)."

- This sentence should be softened given the authors didn't test matched concentrations to PTH1-11.

The transactivation experiments are interesting but some key controls are missing. What is the relative expression of each construct used in Fig. 6, and does co-transfection result in "rescue" of cell-surface expression of one of them (especially the deltaECD construct).

Other minor points

- P1L22, Photons mentioned twice.

- P4L111, ref to Fig1A confusing as Fig1A doesn't mention engineered receptors (1C does)

- P5L162, grammar, "of" missing

Responses to Reviewer Comments for “Highly biased agonism for GPCR ligands via nanobody tethering”.

We appreciate the helpful and extensive feedback provided by reviewers. We have responded to all reviewer comments and have provided extensive new experimental data (14 new Supporting Figures), as described in responses to individual review comments below. These data do not change the core focus of the manuscript, but serve to strengthen conclusions and cover important topics including:

*Evaluation of the mechanistic aspects of PTH1-11-Nb_{PTH1R1} action on wild type PTHR1, such as signaling localization assessed through the addition of a cell-impermeable inhibitor, which rapidly quenches cAMP signaling for PTH1-11-Nb_{PTH1R1} conjugates (Supporting Figure 18). This observation provides further support for the hypothesis that PTH1-11-Nb_{PTH1R1} primarily signals from the cell surface, and that this signaling is persistent unless a competitive antagonist is added.

*Further evaluation of PTH1-11-Nb_{PTH1R1} signaling through the Gq/calcium mobilization pathway (Supporting Figure 21), which confirms previous findings showing that PTH1-11-Nb_{PTH1R1} is highly selective for the Gs/cAMP pathway.

*A demonstration of the generality of the approach described in this manuscript through the development of a new Nb-ligand conjugate that targets GLP1R and shows high potency cAMP induction with minimal bArr-recruitment activity (Supporting Figures 27-28).

*A new assessment of the affinity of Nbs for purified receptor extracellular domains using surface plasmon resonance-based assays demonstrating K_d values in the low nM range (Supporting Figures 14 and 28).

*An evaluation of PTH1-11 folding by circular dichroism, either alone or when linked to Nb_{PTH1R1}, which demonstrates that Nb conjugation does not induce substantial folding (Supporting Figure 26).

REVIEWER COMMENTS (responses in blue)

Reviewer #1 (Remarks to the Author):

Comment: This is an interesting manuscript that describes the development of nanobody-tethered agonists at the PTH1R that display strong Gs bias (with a loss of beta-arrestin and Gq activity (as assessed by calcium)). The authors start by characterizing a nanobody that recognizes the 6E motif and then show the same behavior is true of PTH1-11 linked to nanobodies that target the ECD of the native PTH1R. The binding of those two nanobodies was also evaluated in detail with approaches that included hydroxy radical-based footprinting (not very helpful) and other approaches. There are a considerable number of experiments that are primarily of a technical nature focused on the mode of binding of the agonist, most interesting of which is the experiment which shows that some of this binding can occur in trans across a receptor dimer (although it does not exclude that the primary mode of activity is through a receptor monomer).

I think the studies have been performed well and are rigorous. My main question is regarding the mode of signaling of these conjugates, as the Vilardaga group has shown in a series of manuscripts that prolonged Gs signaling is usually associated with signaling from internalized receptors. This is contrast to this study, where what appears to be prolonged signaling (at least from the kinetic tracings in Fig. 2B) is likely from the plasma membrane due to the lack of barrestin activity and receptor internalization. I think this is an important question that needs to be considered as it would have a significant impact on the predicted biological activity of these compounds, as the Vilardaga group has shown that internalized Gs signaling is what is required for the bone growth response.

Response: We appreciate the thoughtful evaluation from Reviewer 1. The localization of signal generation can be technically tricky to assess but is often evaluated by adding an impermeable inhibitor that preferentially quenches signaling generated from the cell surface. We performed this experiment in the context of engineered receptor (PTHR1-6E; see Supporting Figure 4, previously Supporting Figure 1). We have now added new data with wild-type PTHR1 probed by the addition of the cell-impermeable inhibitor PTHrP(7-36). This inhibitor causes a rapid loss of signaling for NbPTHR1-PTH1-11 relative to PTH1-34 signaling (Supporting Figure 18). This finding supports the hypothesis that NbPTHR1-PTH1-11 induces prolonged signaling from the cell surface, as seen previously for specific analogues of PTH1-34 (see reference Liu et al J. Am. Chem. Soc 2019) and is discussed in the revised text as pasted below.

“Given the rapid decrease of binding as measured by BRET for TMR-NbPTHR1 caused by competitors such as PTHrP7-36 (Supporting Figure 12), we sought to understand how this observation related to the prolonged cAMP signaling responses induced by conjugates such as PTH1-11-NbPTHR1. Building on an assay previously developed to specifically perturb PTHR1 signaling from the cell surface, we assessed the impact of adding a peptide antagonist (PTHrP7 36) during the washout phase of cAMP assays (Supporting Figure 18). We observed that this antagonist had a minimal impact on PTH1-34 washout kinetics, in accord with previous findings. In contrast, the addition of PTHrP7-36 caused a rapid washout of PTH1-11-NbPTHR1 cAMP signaling responses (Supporting Figure 18).”

Comment: Another question is regarding the observed loss of Gq signaling – this was assessing by calcium. Is this also true of the conjugates to the native receptor?

Response: We show analysis of this question in Figure 5E, demonstrating that Nb-PTH1-11 conjugates show very weak or negligible signaling through the Gq/calcium pathway. We have added new more extensive data for calcium mobilization assays to Supporting Figure 21 that support data shown in Figure 5E.

Comment: Lastly, this behavior recalls something seen at chemokine receptors. Chemokines bind use a similar model of binding with an N-terminal binding to an orthosteric site and the globular domain binding the ECD. At CXCR3, there is a splice variant (CXCR3B) of the extracellular domain which results in a longer N-terminus. This change in binding in the ECD results in a receptor that is completely b-arrestin-biased (this has been shown by the Sakmar and Rajagopal labs). Here, it looks like changing the interaction in the ECD results in a Gs-protein-biased agonist.

Response: We thank Reviewer 1 for pointing out this relevant example. We have added citations to this work and text in the discussion section (pasted below, lines 451-454).

“Previous studies with splice variants of CXCR3, in which the extracellular portion is varied, demonstrate that receptor alterations that modify binding mode can lead to pathway specific signaling outcomes [refs PMID 28559424, 27512119]”

Other comments:

Comment: Line 125-126: Should Nb6E-PTH1-11 be PTH1-11-Nb6E?

Response: This typo has now been corrected.

Comment: Table 1: What are the units? I assume nM.

Response: We now include the unit label in this table.

Comment: Figure 6B: The color scheme could be improved to help differentiate between PTH1-11 and PTH1-11-NbPTHR1

Response: We have now changed the color scheme throughout the manuscript.

Reviewer #2 (Remarks to the Author):

Comment: The manuscript entitled “Highly biased agonism for GPCR ligands via nanobody tethering” by Sachdev et al. describes the generation and characterisation of synthetic fusions of nanobodies with the n-terminus of the parathyroid hormone (peptide, PTH) (plus a control where the nanobody is inserted into the parathyroid hormone receptor (PTHR1) and the nanobody epitope is instead fused to PTH n-terminus). These chemical biology derived ligands are then characterized for a subset of their activities in recombinant cells using a variety of different PTH1R constructs.

Response: We appreciate the extensive feedback provided below.

Comment: I have substantial reservations regarding the claims of novelty, data presentation and interpretation, which I detail below. None of the primary contentions of the paper are supported by the data and the argument that the approach provides a facile means to generate biased ligands is not generally true. Many pharmacologically relevant GPCRs have small molecule agonists and little or no extracellular domain, thus an approach using an extracellular directed nanobody (which needs to be developed for every single receptor) coupled to a peptide (only a subset of receptors) does not really fulfill the description of facile or generalizable.

Response: We thank the reviewer for sharing these concerns. While it is true that this approach cannot be extended to all targets, we suggest that many such receptors are amenable to the methods described here including one new example added during revisions (see details below).

*71% of known GPCR ligands peptides or proteins [Pubmed ID **31675498**], indicating the wide applicability of the approach described here.

*In the first draft we demonstrated the generation of biased agonists for three different engineered versions of PTHR1

*We have now added a second example of generation of a biased agonist for a separate receptor (Glucagon-like peptide receptor 1, See Supporting Figures 27-28)

*We provided 8 references for nanobodies that bind to the extracellular face of GPCRs that could be applied to such efforts.

*Given that current efforts to discover biased agonists are empirical, any mechanism-based approaches to preferentially generate biased agonists will be highly valuable.

Comment: There is also no data presented that supports the contention that these ligands will act in a biased manner on an endogenously expressed PTHR1 in a native cell. Thus, even if the data were indisputable, they would be of purely phenomenological value.

Response: While we appreciate the spirit of the reviewer's comment, we emphasize that this is a mechanistic study, with the goal of establishing a precedent for a new mechanism of biased agonism. Biased agonism is an intensely studied area (further discussed below), that stands to benefit from new mechanistic insights, regardless of whether they are ready to be applied in vivo.

The proposed mechanism of action (signaling activation by binding in trans) would be proposed to be dependent on receptor expression levels of both protomers. We show that as predicted from this model, the nanobody-PTH conjugates are inactive on cells expressing very low levels of receptor (Supporting Figure 33). It is important to note that past work demonstrated that Nb-PTH conjugates show biological activity in mice (Cheloha et al Nature Comm 2020), so the lack of activity in cells with low receptor expression levels does not equate to irrelevance in vivo.

Comment: The concept of signal bias at GPCRs has been very popular for at least 25 years, yet despite this there is little evidence that supports physiologically relevant coupling of GPCRs to more than one primary transducer. The failure of Olceridine to provide a substantial improvement in safety window over other opioids should be instructive in the caution with which the GPCR field should have with this concept.

Response: While we appreciate the reviewer's comment, our manuscript is not intended to provide an evaluation of the therapeutic prospects of biased agonism, but rather to report new data that we have identified such an agonist for the PTHR1 using a novel approach. We do not dispute that it remains uncertain whether a direct pursuit of biased agonism for the PTH1R or any other GPCR would be particularly worthwhile in terms of therapeutic approach, but we do recognize that the concept is worthy of exploration and that such studies are a robust area of research, and indeed have provided important insights into mechanisms (see two manuscripts published in 2023 in Nature with biased agonism in their title: <https://www.nature.com/articles/s41586-023-06395-9> and <https://www.nature.com/articles/s41586-023-06467-w>). We nevertheless have added a comment to our Discussion to express such a balanced perspective on the topic generally (in lines 487-489, quoted below).

"Despite this interest, the paucity of pathway selective ligands among approved therapeutics highlights uncertainty about translational prospects."

Comment: The authors claim that their approach, to use sortase mediated ligation of a nanobody with a synthetic peptide, is novel technology (they call this “CLAMP”). This claim is undermined by the fact that sortase mediated protein-protein (and protein-peptide) ligation has been described for at least 19 years: Mao H, Hart SA, Schink A, Pollok BA. Sortase-mediated protein ligation: a new method for protein engineering. J Am Chem Soc. 2004 Mar 10;126(9):2670-1. doi: 10.1021/ja039915e. PMID: 14995162. And is further undermined by the fact that the authors have previously described conjugation of PTH peptides to nanobodies in this journal: Cheloha RW, Fischer FA, Woodham AW, Daley E, Suminski N, Gardella TJ, Ploegh HL. Improved GPCR ligands from nanobody tethering. Nat Commun. 2020 Apr 29;11(1):2087. doi: 10.1038/s41467-020-15884-8. PMID: 32350260; PMCID: PMC7190724.

Response: As the reviewer points out, we make no effort to obscure that this is an approach that was reported by a subset of the authors of this manuscript 3.5 years ago (cited extensively throughout the manuscript). No other research lab in the world has reported using CLAMP, and this evaluation of the conjugates generated using CLAMP for biased agonist properties was new at the time of initial submission. Regardless, we have removed the word novel to avoid any confusion.

Comment: In figure 2 the authors provide data for cAMP responses. This data needs a control of un-modified PTH1R to allow the reader to compare the response of PTH1-34 and 1-11 at this receptor compared to the modified receptors reported. Without this control the data is very difficult to interpret.

Response: We agree that comparison experiments with wild type receptor are important, but we are puzzled that the reviewer was unable to find relevant comparisons in the figure devoted to wild type receptor (Figure 5) . Perhaps we did not make it sufficiently clear that Figure 5 and related supporting figures contain all data corresponding to wild type receptor. We have modified the legend of Figure 5 to indicate that these data correspond to unmodified receptor. We have also added new data (Supporting Figure 2) reporting the action of PTH1-11-6E and PTH1-11-Nb6E on wild type PTHR1.

Comment: The y-axis and figure legend need to be fixed for panels A and D, is this AUC or peak? If AUC this data is inconsistent with panels C and F, where the magnitude of the response is between 5 and 12 times higher after “washout”.

Response: That the reviewer finds this data uninterpretable suggests a problem in the labeling of the graphs, which should be simply interpretable as they show the direct (not normalized) raw luminescence output (as photomultiplier counts per second) vs time, as measured in the plate reader. Washout assays are described in many cited references (ref Nagai 2011, Clark 2020, White 2021, Pena 2022, Liu 2022, Liu 2019, Cheloha 2020, Cabalteja 2022, Maeda 2013) and described in the methods section.

To help clarify these data, and that they are not AUCs, we revised the y-axis label to include the term "cps" and defined cps as luminescence counts per second in the legend. The axes title of cAMP has been adjusted to “cAMP peak”. Washout data are always indicated by AUC labels on the y-axis.

Comment: The data from the “washout” experiments is almost uninterpretable, the EC50 for PTH1-11-Nb6E at PTHR1-6E is not reported, but appears to be in the order of 10 nM (this is reported as being in table 1, which is actually mass-spectrometry data).

Response: We are puzzled by this comment and apologize for the confusion. We request the reviewer consult Table 1 and not Supporting Table 1, where EC50 values are listed for washout assays (where the washout EC50 is 8 nM).

Comment: This ligand appears to be a full agonist, as assessed by panel A, which means one would expect the pEC50 to be to the left of the apparent affinity, since the output being measured is amplified downstream. The affinity of this ligand would therefore be predicted to result in relatively rapid unbinding from the receptor (compare say with figure 4C). It is unclear from the methods section exactly what the protocol is for this “washout” experiment, but it seems that the idea is infinite dilution, in which case the subsidence of signal should reflect ligand unbinding. Since the authors are claiming bias away from beta-arrestin (and therefore, presumable internalisation) this sustained response is not being attributed to a pool of internalised receptors but rather persistence in ligand binding. Minimally the authors need to binding assays to determine the association and dissociation rates of these ligands as well as PTH1-34 at these engineered receptors.

Response: Binding and dissociation kinetics are performed for the wild type receptor in Figure 4. Since engineered receptors are not expressed in physiological settings, we prefer to focus our analysis using wild type receptor.

Comment: The “washout” data in panel E is onconsistent with cAMP data shown above in panel D; in panel D, 1 nM of PTH1-11-6E is saturating for signal, 1 nM PTH1-34 is approximately an EC50 (and less than 50% of PTH1-11-6E max) and PTH11 is a approximate EC80, yet in panel E all 3 of these concentrations start at the same point

Response: The reviewer correctly notes that the magnitude of the cAMP response reported for cAMP (peak cAMP in relabeled graph) does not precisely correspond with the cAMP response measured at the beginning of the washout assay. This is often observed because a buffer exchange and wash is performed between the end of the “peak cAMP” assay and the beginning of the “washout cAMP” assay, providing time for levels of signaling to change. We have modified the methods section text to make this aspect of the protocol more clear (as pasted below).

“After this period, the medium containing ligands was discarded. New CO2 independent medium containing fresh luciferin was added to all wells (resulting in a 2-5 minute interlude between ligand on and washout periods) and luminescence responses was recorded for an additional 30 min (ligand-off phase).”

Comment: Accompanying this figure is Supp Figure 1, where the authors appear to report that in a “washout” (infinite dilution??) experiment a competitive inhibitor causes the cAMP signal to decrease more rapidly. This is inconsistent with the general model for binding via mass action, where a competitive antagonist would NOT be expected to change the unbinding rate of an agonist. This needs to be explained.

Response: Although we do not have a full explanation for the mechanism the underlies this behavior, we provided a speculative interpretation in the text (lines 150-156 in original, line 156-

160 in revised): “To assess whether the prolonged signaling from PTH1-11-6E corresponded with continuous engagement of receptor, or whether two site binding permitted repeated association/dissociation cycles at different receptor sites, we evaluated the impact of antagonists added at the beginning of the washout phase (Supporting Figure 1). PTH1-11-6E washout responses were highly sensitive to inhibitors. Addition of synthetic 6E peptide causes inhibition through competition with PTH1-11-6E for Nb6E binding, whereas SW106 interferes with the binding of PTH1-11³³. The marked effects of both types of inhibitors on PTH1-11-6E signaling suggests that this ligand does not continuously engage both of its binding sites throughout washout.”

To address this mechanistic question further with experiment (using wild type PTHR1) we have added new experimental data in which we perform a washout study wherein an antagonist (PTHrP7-36) that binds to the PTHR1 ECD is added during the washout phase (Supporting Figure 18). Past work (Liu, JACS 2019) has shown that this antagonist facilitates a more rapid “washout” of a conventional peptide ligand that preferentially signals from PTHR1 found at the cell surface. As shown in new data we see that PTHrP7-36 induces a more rapid loss of signaling for Nb-PTH1-11 conjugates than for PTH1-34, which is known to signal from receptors located within endosomal compartments (that are inaccessible to PTHrP7-36). We have added new text to the manuscript discussing this observation (lines 287-298 in revised):

“Building on an assay previously developed to specifically perturb PTHR1 signaling from the cell surface²⁵, we assessed the impact of adding a peptide antagonist (PTHrP7 36) during the washout phase of cAMP assays (Supporting Figure 18). We observed that this antagonist had a minimal impact on PTH1-34 washout kinetics, in accord with previous findings. In contrast, the addition of PTHrP7-36 caused a rapid washout of PTH1-11-NbPTHR1 cAMP signaling responses (Supporting Figure 18). This observation parallels findings with PTH1-11-6E peptides (Supporting Figure 4) and suggests that PTH1-11-NbPTHR1 conjugates signal mostly from the cell surface and may engage in cycles of binding, dissociation, and reassociation enabled by a multi-site binding mechanism. Another non-exclusive possibility is that PTHrP7-36 acts allosterically to induce PTH1-11-NbPTHR1 dissociation, potentially explaining the discrepancy between the high affinity binding recorded by surface plasmon resonance (Supporting Figure 14) and the rapid rate of cAMP signal washout for PTH1-11-NbPTHR1 upon addition of PTHrP7 36.”

Comment: The authors should not present representative data from 1 experiment, they must present data using the mean values of the technical replicates from separate independent experiments.

Response: We stress that all cAMP data is shown in unprocessed (un-normalized) form to enable direct evaluation of assay data with minimal processing. Variation in the amplitudes of raw signals measured in experiments performed on different days does not allow for averaging across experiments performed on days.

To facilitate comparison between experiments performed on different days, all data is analyzed by conventional dose-response assays and compared to an internal control ligand (PTH1-34) ensuring that assays run day-to-day are comparable. All replicate data is either shown as primary data in the supporting information or is represented in the quantitative data shown in table 1. To ensure full access to data for inspection we have now inserted dose-response graphs for all independent replicates into newly added Supporting Figures 1, 3 and 17.

Comment: In the legend the curve-fit is reported as being from a 3-parameter logistic equation, this is clearly not true, the hill slope is clearly floated for PTH1-11 (panels A & D) and PTH1-11-6E (panel D) and may be floated for other panels. At present this data is not suitable for publication.

Response: We re-assessed the graphics and analysis used and discovered that graphics mistakenly used a 4-parameter curve fitting, whereas data analysis used a 3-parameter fitting. Graphics have been corrected to show a 3-parameter fitting.

Comment: Figure 3. Again, these assays need a control of un-modified PTHR1 to allow assessment of data.

Response: As noted above, we have worked to emphasize that all data with un-modified PTHR1 can be found in Figure 5.

Comment: The data in panel A is inconsistent with cAMP data presented in figure 1. As the authors correctly identify, cAMP is an amplified downstream signalling measure. The potency for cAMP should be (minimally) equivalent to that for Galphas recruitment but more typically to the left. This is not the case, and the authors do not address this apparent inconsistency.

Response: Potencies can be assay dependent and inconsistent with what is expected based on biophysical expectations. The potencies observed for PTH1-34 in activation of PTHR1 Gs recruitment vs induction of cAMP production is consistent with previous reports (for Gs-BRET on PTHR1 see Avet et al eLife 2022—EC50 ~0.1 nM vs. cAMP Glosensor assay for PTHR1 EC50 ~ 0.5-50 nM see <https://www.nature.com/articles/s41467-022-34009-x> for reports from authors independent of this work). We have now directly mentioned this past work for comparison in the manuscript text. We suggest that it is beyond the scope of this manuscript to interrogate mechanistic details of previously described assays unless there is a specific connection to the main thrust of the work.

Comment: Calcium data needs to minimally be reported as concentration-response curves and preferably representative kinetic traces should be shown (at least in supplementary). Calcium mobilisation is typically downstream of Galphaq, but is known also to be downstream of other transducers, without the use of a specific Galphaq inhibitor, this data could have alternative interpretations. Data is again shown as mean and error of technical replicates, this is not appropriate.

Response: We have now added dose-response data for calcium responses for in Supporting Figure 21. In this figure we have also shown that addition of an inhibitor of Gq function blocks PTH mediated-calcium mobilization. This calcium data shown in main figures is shown as mean and error of biological replicates, not technical replicates.

Comment: While the use of *, ** etc. to indicate different thresholds of statistical significance is common practice, however it demonstrates a misunderstanding of statistics and should be strongly discouraged.

Response: We now follow conventions specified by the Journal, with exact p-values quantified where listed.

Comment: Figure 4. Panel A requires specificity controls, all ligands need to be tested against

cells expressing an irrelevant receptor (e.g. another Class B GPCR). For ligand binding assays data needs to be provided that demonstrates the equivalent function of nLuc labelled PTH1R with unlabelled receptor and all assays require a specificity control of either a nLuc labelled irrelevant receptor OR an excess of unlabelled ligand to establish un-specific binding.

Response: Previously published and cited work (Yu 2022) has demonstrated the equivalence of PTHR1 and PTHR1-nLuc for signaling. We suggest that reproducing previously published controls will not provide new information or increased reliability of this data.

We provide the requested specificity control (unlabeled Nb_{PTHR1} added to Nb_{PTHR1}-TMR) in Supporting Figure 12D. We have also added new data showing that Nb_{PTHR1} does not bind to a cell line expressing GLP1R instead of PTHR1 (Supporting Figure 28).

Comment: Panel B is data from a binding assay using the nLuc tagged PTH1R, the figure legend needs to be rewritten to make this clear.

Response: We have now ensured that Figure 4B shows the label for nLuc PTHR1 both in the figure legend and on the y-axis of the graphic.

Comment: Since this is a saturation binding assay there needs to be a specificity control (above). It is also more usual to display this type of data on a linear scale with a hyperbolic curve and to fit to a saturation binding equation (taking into account unspecific binding).

Response: We have provided specificity controls in Supporting Figure 12. This manuscript is not the first report of this assay system (see Yu 2022). Our data here with the saturation binding of PTH1-34-TMR only serves to reinforce the reproducibility of previously published data. We have reformatted Figure 4b to show a linear x-axis.

Comment: In the text, figure 5B is referred to as cAMP data (line 274, page 10), which clearly it is not and I do not see cAMP data for these constructs elsewhere in the manuscript.

Response: We are puzzled by this comment, as Figure 5b does show cAMP data.

Comment: Panel C needs specificity control. The 2 μ M PTH1-34 addition makes two issues very evident; firstly there is some unspecific binding of the probe, and second the unbinding rate of the NbPTHR1-TMR ligand is extremely fast (consistent with the estimated low affinity of \sim 500 nM and the fast Kobs shown in supp fig 7.) – this is extremely difficult to reconcile with the “washout” data shown elsewhere in the paper – particularly figure 5, where the ligands would be expected to almost immediately unbind from the receptor.

Response: We direct the Reviewer to Supporting Figure 12d for a specificity control for NbPTHR1-TMR.

The divergence between the binding (dissociation) kinetics observed in the BRET binding assay and the duration of the cAMP response observed upon washout is indeed interesting. We suggest that expected (rapid) washout behavior (based on washout in BRET binding assays) is observed upon addition of a competitive antagonist for an engineered receptor-nanobody system (See Supporting Figure 4). We have added new data showing that the same rapid washout is observed on wild type PTHR1 upon addition of a competitor (Supporting Figure 18). Thus, we hypothesize that some aspect of the two-site binding behavior of Nb-PTH1-11 conjugates facilitates prolonged signaling in the absence of a competitor, but is rapidly

quenched by the addition of a competitive antagonist. This is now discussed in the manuscript text (see response to comment above).

We have now also performed an independent assessment of Nb-PTHr1 extracellular domain binding (Supporting Figure 14). This new data shows that NbPTHr1 binds to purified PTHr1-ECD with a K_d of <10 nM, consistent with the prolonged washout data observed for NbPTHr1-PTH1-11 conjugates.

Comment: The competition experiments shown in the bar-graph (no label but next to panel C) have no indication either in the method nor figure legend of the concentration of competitor, the reader has to go to supp fig 9 to find concentrations, please add concentrations to figure legend.

Response: We apologize for the confusing formatting. Both graphics in panel c are part of the same panel. This is now noted in the Figure legend. We have added concentrations of competitors to the figure legend.

Comment: Supp fig 9 data is very problematic; 3 concentrations of NdPTHr1-TMR ligand are used in each competition experiment with a high degree of variability in both window and apparent affinity. For example the maximum signal in panel C is ~0.028, whereas this is 0.06 in panel F (Supp fig 9). More concerning is that the apparent probe affinity varies between experiments, panel C appears to have at least a 10 fold, but more likely a 30 fold higher affinity for the probe ligand compared with panel E (Supp fig 9).

Response: We appreciate the reviewer's careful observation, but suggest that interpretation of these data in this way is beyond the experimental question being addressed. Data in Supporting Figure 12 (formerly Supporting Figure 9) is not intended to assess ligand affinity, but rather whether two ligands bind to the same site. The issues raised do not call into doubt the conclusions regarding shared binding epitopes, corroborated using many other assays. More technically, these are simply the observed levels of experimental variability in this assay run in this mode, presented without in-experiment normalization.

Comment: In D, why were more concentrations of unlabelled competitor included and the data fitted to a competition binding model to enable estimation of affinity?

Response: We note that BRET-based binding assay has not been validated to be run in this mode to determine affinity. We also emphasize that Supporting Figure 13 shows a dose-response competition assay, which shows comparable performance of PTH1-34 and NbPTHr1 for competing with PTH1-34 for binding, suggesting a similar affinity for the two compounds.

Also relevant to these discussions, we have added new data showing an alternative estimate of binding through surface plasmon resonance-based assays (Supporting Figure 14).

Comment: Figure 5. "Washout" phase data is inconsistent with binding data, this needs to be addressed (see above).

Response: Please see the response above addressing differences in washout assays either with or without competitive antagonists added during the washout phase. We further stress that our findings with washout data were consistent and robust with different receptor variants.

Comment: Calcium data needs to be presented as concentration-response.

Response: We have added new data for calcium mobilization assays in Supporting Figure 21.

Comment: Representative data needs to be removed and complete, pooled data set from means of individual experiments needs to be reported.

Response: As discussed above, the magnitude of signals in cAMP (peak and washout) varies from day-to-day, whereas dose-response curves are comparable. We prefer not to modify data by normalizing responses. This necessitates using the dose-response parameters to compare replicates. These data are described in Table 1 and all replicate experiments have now been added to Supporting Figures 1, 3, and 17.

Comment: Figure 6. Why is the magnitude of the response so much lower than all other assays (less than 1/6th by my estimation)? There is not an appropriate control for this assay set. Since the responses are so much lower than all other reported data it is feasible that the rPTH1R1-null is capable of signalling at substantially reduced levels, which would mean all the signalling being measured is simply happening in-cis rather than in-trans. A control where this rPTH1R1-null receptor is tested alone is required for interpretation of the data. Representative graphs need to be removed and complete, pooled data set from means of individual experiments need to be reported.

Response: We have added new data to assess the signaling of cells transfected only with rPTH1R1-null (data in Supporting Figure 30). We observe signal responses to PTH1-34 that do not rise above baseline upon transfection with rPTH1R1-null.

Regarding the level of signaling observed in this Figure, this cell line expresses a different receptor at a different level than previous assays and different levels of signaling are not unexpected.

We have now added all replicates for data in Figure 6 into Supporting Figure 31.

Minor comments:

Comment: Please provide the LC-MS traces for all compounds so that the reader can assess purity. (supplementary tables 2 and 2).

Response: We have added mass spectrometry data from LC/MS for all compounds as a Supporting File.

Comment: Please provide supplementary data using a cell-permeable cAMP analogue (e.g. 8-CPT) to ensure the Glowsensor cAMP biosensor is not saturated at maximal PTHR1 ligand concentrations.

Response: Glowsensor has been established as an assay for measuring cAMP responses for over 10 years. It is beyond the scope of this manuscript to perform mechanistic studies of well-established assays. For the reviewer's reference we include example data (for review only) showing that forskolin induces an additional cAMP response reflected by additional luminescence signal even after pretreatment with a saturating concentration of PTH1-34 in cells expressing PTHR1.

Supporting Figure for Review: Data points correspond to mean \pm SEM from 12 technical replicates in a single experiment.

Comment: EC50 data is shown with linear SEM, presumably the data is log-normally distributed?

Response: We do not claim to show that enough data to conclude that any individual data set is log-normally distributed. We have now included all primary data for main figures (Supporting Figures 1, 3, 17, 31) for readers to assess any desired data set.

Reviewer #3 (Remarks to the Author):

Comment: The manuscript by Sachdev et al (Highly biased agonism for GPCR ligands 1 via nanobody tethering) presents interesting pharmacological observations around linking the "activating" N-term sequence of PTH peptide to PTHR1 ectodomain-binding nanobodies (Nbs). In full length PTH peptide, it is well established that the C-term peptide sequence binds to the PTHR1 ectodomain with high affinity, thus allowing the weaker affinity N-term sequence to engage with the PTHR1 transmembrane domains and activate the receptor. Sachdev et al replaced the C-term PTH sequence with a range of Nbs that could either bind to engineered epitope tags in PTHR1, or natural epitopes in PTHR1. In each case, Nb fusion resulted in improved potency of the activating peptide at PTHR1 and a perceived loss of beta-arrestin efficacy compared to the activating peptide alone. These data shed light on the mechanisms by which PTH binds and activates PTHR1, and perhaps a means to engineering interesting agonists at Class B GPCRs, and thus is worthy of publication. The authors conclusions on mechanism, however, are partially flawed and should be modified by addressing the concerns below.

Response: We appreciate the thoughtful comments from Reviewer 3.

Comment: Kobayashi et al (doi.org/10.1016/j.molcel.2022.07.003) identified that PTH retains significant dynamics when bound to PTHR, and that these dynamics (especially partial activating peptide sequence dissociation) play a key role in G protein activation. The authors should discuss the relevance of their data to Kobayashi's findings, since Nb fusion would certainly change the dynamics of PTH1-11 when bound to PTHR1.

Response: We agree with the Reviewer that the dynamics of PTHR1 bound ligands is an exciting area of ongoing investigation. We have now added new text and a new citation to the discussion section to describe this previous work on PTHR1-ligand structure and dynamics in the context of the findings of this manuscript (pasted below, line 624-626).

“We speculate that PTH1-11-Nb might engage different dynamics for receptor conformational changes and activation in comparison to those recently described for natural ligands of PTHR1⁴⁷.”

Comment: What is known about the folding of PTH1-11 and the influence of Nb fusion on alpha helix formation? It is becoming apparent for similar peptide receptors that peptide folding is influenced by receptor binding, and that peptide folding dynamics are important for receptor efficacy and signaling bias. The authors could perform CD spectroscopy or NMR to reveal any influence of Nb fusion on PTH1-11 folding.

Response: We have now performed experimentation to evaluate the folding of PTH1-11 either in isolation or upon linkage with Nb_{PTHR1} (Supporting Figure 26). These experiments show that PTH1-11 does not exhibit a circular dichroism signal consistent with helix formation either alone or when linked to Nb_{PTHR1}. It is possible that Nb binding to receptor helps to facilitate the folding of PTH1-11 (in the context of Nb-PTH1-11 conjugate), analogous to the contribution of PTH12-34 binding on PTH1-11 in the context of PTH1-34. These enticing studies constitute an entirely different set of experimental techniques and are beyond the scope of this manuscript. We have pasted newly added discussion of this data below.

“Circular dichroism (CD) analysis of PTH1-11, NbPTHR1, and PTH1-11-NbPTHR1 conjugate demonstrated that PTH1-11 adopts primarily a random coil confirmation, which is not dramatically changed upon conjugation with NbPTHR1 (Supporting Figure 26). The sum of CD readings recorded for PTH1-11 and NbPTHR1 is similar to the CD spectrum recorded for PTH1-11-NbPTHR1 conjugate, suggesting that neither building block substantially affects the conformational propensities of the partner component when conjugated.”

Comment: The complete lack of efficacy of PTH1-11-Nbneg is a concern. Shouldn't this fusion have similar efficacy to PTH1-11 alone? Have the authors tried a range of non-binding Nbs? Again, analysing the influence of Nb fusion on PTH1-11 fold may help reveal what happened to this control.

Response: The origin of the lack of activity of PTH1-11-Nbneg is unknown but it has been documented for many PTH1-11-Nb conjugates in which the Nb does not bind to a target expressed on the cell surface (see Cheloha et al Nat Comm 2020). We have added a new supporting figure to show replication of this finding this for another Nb-PTH1-11 conjugate (new Supporting Figure 2) that does not involve Nbneg.

Comment: PTH1-11 seems to be less efficacious on PTHR1-6E compared to PTHR1-Nb6E, especially at beta-arrestin. Any ideas why?

Response: We observe somewhat variable efficacy for PTH1-11 in inducing beta-arrestin recruitment for PTHR1-6E and PTHR1-Nb6E. We have added graphs for more independent

replicates in new Supporting Figures 1 and 3). In viewing this data in total, there does not appear to be a reproducible difference in the efficacy of PTH1-11 for beta-arrestin recruitment at these receptors.

Comment: The authors also need to test PTH1-11-Nb6E and PTH1-11-Nbneg at wtPTHR1 at all pathways to be able to properly comment on their findings (i.e. are they also biased at the wt receptor?).

Response: We have now added data to Supporting Figure 2 showing that each of these conjugates is inactive for signaling through wild type PTHR1.

Comment: The insertion point of the 6E epitope was proposed to be unstructured in the receptor, yet the authors identified that NbPTHRX2 likely binds to this site. Does the fact that a Nb was identified that binds to this region suggest that it may have indeed have fold?

Response: Antibodies that bind to conformational (discontinuous) epitopes (those formed upon proper folding of the protein) typically require that the antigen adopt a specific fold to enable antibody binding. In this work, NbPTHRX2 is shown to bind to a linear epitope (see blockade of Nb binding by synthetic peptide in Supporting Figure 8). Linear epitopes do not require folding to enable facilitate binding with antibodies (or nanobodies), so it is still possible that this region of the receptor is unstructured. We have now emphasized this subtle point and its implications in the manuscript text as pasted below.

“The effective blockade of NbPTHR-X2 binding to receptor by a short peptide suggests that epitope folding is likely not a major determinant of Nb recognition. This aligns with previous work demonstrating that the portion of PTHR1 encompassing this epitope is dynamic and not easily characterized in structural studies”

Comment: Fig 5D, PTH1-11-NbPTHR1 concentration was too low to see whether this molecule could act at Beta-arr, why? If the authors could have gone as high as PTH1-11, what would they expect?

Response: This is a reasonable question but difficult to answer with certainty. The highest concentration tested in Figure 5D is 300 nM of PTH1-11-NbPTHR1. In assessing binding of this conjugate in flow cytometry (Supporting Figure 25) we see very strong staining of cells at 300 nM, suggesting a substantial portion of receptor is occupied with PTH1-11-NbPTHR1. This suggests that using higher concentrations of PTH1-11-NbPTHR1 (which is not technically feasible due to solubility and stability considerations) would likely not lead to beta arrestin recruitment. We have now commented on this possibility in the main text (pasted below, line 347-352).

“Due to solubility constraints, we were unable assess concentrations of PTH1-11-Nb conjugates above 300 nM, so we cannot exclude the possibility that higher concentrations would induce b-arrestin recruitment. However, we note that PTH1-11-NbPTHR1 induced maximal cAMP responses at concentrations below 10 nM (Figure 5) and that PTH1-11-6E did not induce arrestin recruitment at concentrations up to 10 μ M (Figure 3B).”

Comment: - P9L279 "Analogously to experiments with engineered receptors, both NbPTHR1- and NbPTHR1-X2-PTH1-11 conjugates displayed negligible recruitment of β -arrestin 2 to the plasma membrane or early endosomes in cells stably expressing hPTH1 (Fig. 5D-E, Supporting Figure 12)."

- This sentence should be softened given the authors didn't test matched concentrations to PTH1-11.

Response: We have modified the text to reflect this accurate critique (see previous comment).

Comment: The transactivation experiments are interesting but some key controls are missing. What is the relative expression of each construct used in Fig. 6, and does co-transfection result in "rescue" of cell-surface expression of one of them (especially the deltaECD construct).

Response: We have added new flow cytometry data showing co-expression of rPTH1-null with YFP-delECD-PTH1 does not lead to a change in the level of YFP-delECD-PTH1 expression (Supporting Figure 32). The flow cytometry experiments also demonstrate that YFP-delECD-PTH1 is expressed at a higher level than rPTH1-null upon co-expression (Compare flow cytometry staining levels in Supporting Figure 32 to Supporting Figure 30).

Other minor points

Comment: - P1L22, Photons mentioned twice.

Response: A reference to photons has been removed.

Comment: - P4L111, ref to Fig1A confusing as Fig1A doesn't mention engineered receptors (1C does)

Response: We now refer to Figure 1 instead of Fig 1A.

Comment: - P5L162, grammar, "of" missing

Response: This has been corrected.

REVIEWERS' COMMENTS

Reviewer #1 (Remarks to the Author):

The authors have addressed my comments. I do not have any significant concerns.

Reviewer #2 (Remarks to the Author):

The authors have made a genuine attempt to address my concerns with the manuscript. I appreciate the additional data, however there are still some internal inconsistencies that I find troubling, which have been (largely) argued around in rebuttal. The most serious of my original concerns remain. That is, that the novelty, physiological relevance and general applicability are still not there.

Broad applicability of the approach:

Perhaps I've misread PMID:31675498, but in the reported GPCRs (non-olfactory) there appear to be 55% that are claimed to be peptide binding. Thus, if we include olfactory GPCRs the best estimate would be ~27% peptide binding. A range of very important non-peptide GPCRs, which are current therapeutic targets, would not be accessible. The reach claim (see below also) is that biased ligands generated through this process could be useful therapeutically. If we set aside the issue of generating such ligands for non-peptide GPCRs, such ligands (i.e. peptide-nanobody fusions) will be very unlikely to be centrally penetrant, so many clinically important GPCR targets will be inaccessible to this approach. In an effort to demonstrate better generalizability the authors generate a GLP-1R nanobody:peptide fusion. While this is commendable, this is another Class B GPCR (of which there are only 15 in the human genome) possessing a very large extracellular domain. To properly demonstrate generalizability I would have liked to have seen this done with a class A peptide receptor, perhaps muOpioid, angiotensin II, oxytocin, vasopressin, GHSR1a, or other clinically relevant target.

Physiological relevance:

No attempt has been made to demonstrate bias of these ligands in native tissue. No experiments in animal models are included.

Reviewer #3 (Remarks to the Author):

The authors have addressed my concerns.

Reviewer #2 (Remarks to the Author):

Comment: The authors have made a genuine attempt to address my concerns with the manuscript. I appreciate the additional data, however there are still some internal inconsistencies that I find troubling, which have been (largely) argued around in rebuttal. The most serious of my original concerns remain. That is, that the novelty, physiological relevance and general applicability are still not there.

Response: We appreciate the reviewer's amended comments and acknowledgment of the newly incorporated results.

Comment:

Broad applicability of the approach:

Perhaps I've misread PMID:31675498, but in the reported GPCRs (non-olfactory) there appear to be 55% that are claimed to be peptide binding. Thus, if we include olfactory GPCRs the best estimate would be ~27% peptide binding. A range of very important non-peptide GPCRs, which are current therapeutic targets, would not be accessible.

Response: We appreciate the reviewer's careful assessment of the cited reference. The percentage we described in our last correspondence was derived from an updated analysis presented by the corresponding author of PMID 31675498 (unpublished work). We agree with the reviewer's assessment that the cited work identifies 55% of analyzed GPCRs that bind to peptide or protein ligands, although this assessment does not include olfactory and taste receptors. To emphasize the limitations of the current approach we have added a relevant sentence to the discussion: "Broad application for targeting the GPCR superfamily will require extension of this approach to small molecule ligands."

Comment: The reach claim (see below also) is that biased ligands generated through this process could be useful therapeutically. If we set aside the issue of generating such ligands for non-peptide GPCRs, such ligands (i.e. peptide-nanobody fusions) will be very unlikely to be centrally penetrant, so many clinically important GPCR targets will be inaccessible to this approach. In an effort to demonstrate better generalizability the authors generate a GLP-1R nanobody:peptide fusion. While this is commendable, this is another Class B GPCR (of which there are only 15 in the human genome) possessing a very large extracellular domain. To properly demonstrate generalizability I would have liked to have seen this done with a class A peptide receptor, perhaps muOpioid, angiotensin II, oxytocin, vasopressin, GHSR1a, or other clinically relevant target.

Response: We agree that targeting other GPCRs using this approach will be important to demonstrate generalizability and that such studies are currently underway.

Comment:

Physiological relevance:

No attempt has been made to demonstrate bias of these ligands in native tissue. No experiments in animal models are included.

Response: The reviewer's comment is accurate and was acknowledged previously. Future experiments will seek to translate these findings to tissue samples and animal studies.